# Forces during cellular uptake of viruses and nanoparticles at the ventral side

Tina Wiegand [1,2,3]*, Marta Fratini[1,2,4,5,6], Felix Frey[7,8], Klaus Yserentant [2,7], Yang Liu [9,10], Eva Weber[1,2,11], Kornelia Galior[9,12], Julia Ohmes[1,2,13], Felix Braun[2,7], Dirk-Peter Herten [2,7,14], Steeve Boulant[4,5], Ulrich S. Schwarz [7,8], Khalid Salaita [9], E. Ada Cavalcanti-Adam [1,2]* & Joachim P. Spatz [1,2]*

Many intracellular pathogens, such as mammalian reovirus, mimic extracellular matrix motifs to specifically interact with the host membrane. Whether and how cell-matrix interactions influence virus particle uptake is unknown, as it is usually studied from the dorsal side. Here we show that the forces exerted at the ventral side of adherent cells during reovirus uptake exceed the binding strength of biotin-neutravidin anchoring viruses to a biofunctionalized substrate. Analysis of virus dissociation kinetics using the Bell model revealed mean forces higher than 30 pN per virus, preferentially applied in the cell periphery where close matrix contacts form. Utilizing 100 nm-sized nanoparticles decorated with integrin adhesion motifs, we demonstrate that the uptake forces scale with the adhesion energy, while actin/myosin inhibitions strongly reduce the uptake frequency, but not uptake kinetics. We hypothesize that particle adhesion and the push by the substrate provide the main driving forces for uptake.

[1] Max Planck Institute for Medical Research, Jahnstraße 29, 69120 Heidelberg, Germany. [2] Institute for Physical Chemistry, Heidelberg University, INF 253, 69120 Heidelberg, Germany. [3] Max Planck Institute of Molecular Cell Biology and Genetics, Pfotenhauerstraße 108, 01307 Dresden, Germany. [4] Department of Infectious Diseases, Virology, University Hospital, INF 324, 69120 Heidelberg, Germany. [5] German Cancer Research Center (DKFZ), INF 581, 69120 Heidelberg, Germany. [6] Department of Cellular Biochemistry, Institute of Biochemistry and Biotechnology, Martin-Luther-University Halle-Wittenberg, Kurt-Mothes-Str. 3a, 06120 Halle (Saale), Germany. [7] BioQuant Center, Heidelberg University, INF 267, 69120 Heidelberg, Germany. [8] Institute for Theoretical Physics, Heidelberg University, Philosophenweg 19, 69120 Heidelberg, Germany. [9] Department of Chemistry, Emory University, 1515 Dickey Drive, Atlanta, GA 30322, USA. [10] Johns Hopkins University, 3400N Charles St, Baltimore, MD 21218, USA. [11] Department of Neuroscience, Carl von Ossietzky University Oldenburg, Carl-von-Ossietzky-Straße 9-11, 26129 Oldenburg, Germany. [12] Department of Pathology and Laboratory Medicine, University of Wisconsin School of Medicine and Public Health, 600 Highland Ave, Madison, WI 53792, USA. [13] Experimental Trauma Surgery, Universty Hospital Schleswig-Holstein, Arnold-Heller-Straße 3, 24105 Kiel, Germany. [14] Institute of Cardiovascular Sciences & School of Chemistry, Medical School, University of Birmingham, Edgbaston B15 2TT, UK. *email: wiegand@mpi-cbg.de; elisabetta.cavalcanti-adam@mr.mpg.de; spatz@mr.mpg.de

In vivo, polarized epithelia provide a robust barrier, which pathogens, such as viruses, typically encounter from the apical side. However, infections or microwounds caused by physical or chemical damage can disrupt the epithelial integrity and, as it was first observed for reoviruses, intestinal M-cells can transcytose pathogens through the epithelium barrier[1]. This allows virions to gain access to the basal side, which displays a different subset of cellular receptors triggering viral uptake. Moreover, viruses can get trapped in close contact to cells by interaction with the extracellular matrix (ECM). This can dramatically increase pathogen chances for cellular uptake, for example by providing a push into the ventral membrane, and leading to infection by influencing the mechanical context in which particles are presented to cells[2]. In fact, reoviruses have been found to preferentially infect the basolateral surface of human respiratory epithelial cells[3] and human papillomavirus bind basal ECM components, like shed syndecans, heparan sulfate and laminin 332, preceding entry into keratinocytes[4–6]. Furthermore, a variety of viruses has been identified to mimic ECM motifs to interact with host cells and initiate their uptake[7,8]. Reovirus outer capsid protein λ2 contains the conserved integrin-binding sequences Arg-Gly-Asp (RGD) and Lys-Gly-Glu (KGE)[9]. While primary attachment to host cells was shown to be independent of integrins, these receptors play a role for functional virus entry, possibly through recruitment of the clathrin machinery[10]. Here we propose that integrins have an additional contribution based on force transduction, which could act on virus particles and facilitate their uptake.

Measurements of the forces exerted on single virus particles during endocytosis with atomic force microscopy[11–17] or optical tweezers[18,19] are limited to the dorsal side of adherent cells. Here we develop an experimental approach based on single molecule tension sensors to reveal the minimal forces exerted on single virus particles presented to cells from the ventral side in combination with a matrix mimicking background. Next, we couple viruses and nanoparticles non-covalently via biotin-neutravidin to the substrate and analyze their uptake kinetics as a measure of force by applying Bell's equation. Integrin-specific adhesion between nanoparticles and cells increases the chance of uptake and leads to higher uptake forces. However, even nanoparticles without specific binding motifs are internalized, suggesting, that the deformation of the membrane alone might be sufficient to initiate uptake which is further corroborated by the finding that more particles get internalized where close cell-matrix contacts form. A mathematical model predicts a linear relation between uptake force and adhesion energy and reveals an adhesion energy density for reovirus ($0.18$ mJ m$^{-2}$) that is approximately twice as large as the non-specific adhesion of nanoparticles ($0.10$ mJ m$^{-2}$). This work provides a general approach for mapping the mechanical interaction of virus particles and nanoparticles at the ventral side of host cells. Future work is required to identify the specific mechanisms underlying force-mediated recruitment of cellular structures involved in particle uptake.

## Results

**Design of molecular tension sensors and virus coupling.** To visualize forces exerted on single viruses by adherent cells, we adapted our extracellular tension probes consisting of a surface-bound gold nanoparticle (AuNP) and the fluorescently labeled protein domain I27 of titin[20], which unfolds upon force indicated by an increase in fluorescence (Fig. 1a). The AuNPs thereby serve as (i) distance-dependent fluorescence quencher of the Alexa647 labels through nanometal surface energy transfer (NSET), and (ii) immobilization sites for the I27-tension probes via two cysteine residues at their C-terminus. On the N-terminus the unnatural

amino acid p-azidophenylalanine allows for covalent tethering of different types of alkyne-bearing ligands via copper(I)-catalyzed alkyne-azide cycloaddition (CuAAC) known as click chemistry (see the "Methods" section). To validate that labeling of the I27-tension probe and coupling of specific ligands does not affect ligand binding to cell receptors, we monitored integrin-mediated forces exerted upon binding to the peptide RGD-alkyne (Supplementary Fig. 1). To quantify the forces occurring during virus uptake, we clicked mammalian reoviruses after bioconjugation of alkyne-linkers to their capsids to the surface immobilized Alexa647-I27 tension sensors. We confirmed that single virus particles are covalently immobilized by scanning electron microscopy (SEM) (Fig. 1b) and fluorescence microscopy (Supplementary Fig. 2). We recently showed that performing capsid modifications and click chemistry neither alters reovirus diameter of ~80 nm, nor hinders particle uptake and viral infectivity[21]. To mimic the ECM, the cell adhesion promoting peptide cRGD was co-presented by covalent immobilization on the passivating polyethylene glycol (PEG)-layer, thus allowing cells to adhere and spread on the substrates[22].

**Monitoring of forces on viruses at the ventral cell side.** To observe forces exerted by cells on reoviruses immobilized via the titin-based tension sensors, we performed time-lapse total internal reflection fluorescence (TIRF) imaging 1 h post seeding cells (Fig. 1c and Supplementary Movie 1). Note that viruses were sparsely labeled with Alexa568, to minimize spectral bleed-through and interference with viral function; the sparse labeling however, was accompanied by substantial photobleaching. Fluorescence intensity traces were analyzed at sites of virus immobilization after correcting for the local background (see the "Methods" section). Within 10 min of recording, we could observe fluorescent burst events, indicating that cells exert forces on the viral particles that exceed the unfolding force of the titin-based tension probes. We normalized the frequency of fluorescent events associated with virus particles to the total number of viruses under each cell (70–300 total particles per cell) and divided it by the observation time. This ensemble averaging allowed us to obtain the rate constant $k_{unfold}$ of unfolding events per virus (Fig. 1d). In our experiments, background signals may occur due to impurities, blinking of single molecule tension probes or thermal fluctuations inducing force-independent unfolding. However, these events in the control region outside of cells or at random spots under the cell (not co-localizing with virus particles) were significantly less abundant (Fig. 1d). Subtracting the mean rate of events occurring at virus particles in the control region (including the unfolding at zero-force) from the unfolding rate under cells defined the apparent unfolding rate constant (±s.e.m.):

$$k_{unfold} - k_{unfold}^0 = 1.3\,(\pm 0.2) \times 10^{-4}\,\text{s}^{-1} \qquad (1)$$

Due to this low frequency, we only detected single events per virus during the observation time and did not consider the duration of the fluorescent bursts, which is limited by refolding of the tension probe and photobleaching of the attached fluorophore.

We reasoned that the difference between the unfolding rates underneath and outside the cell area is caused mainly by forces acting at the ventral side of the cell. Such forces might result from direct pulling by the cell, or from shearing motion on the particles trapped between cell and substrate. Irrespective of its origin and direction, any physical force acting on the bond should increase its unfolding rate. Bell's model makes this notion quantitative by assuming that an applied force lowers the energy barrier for

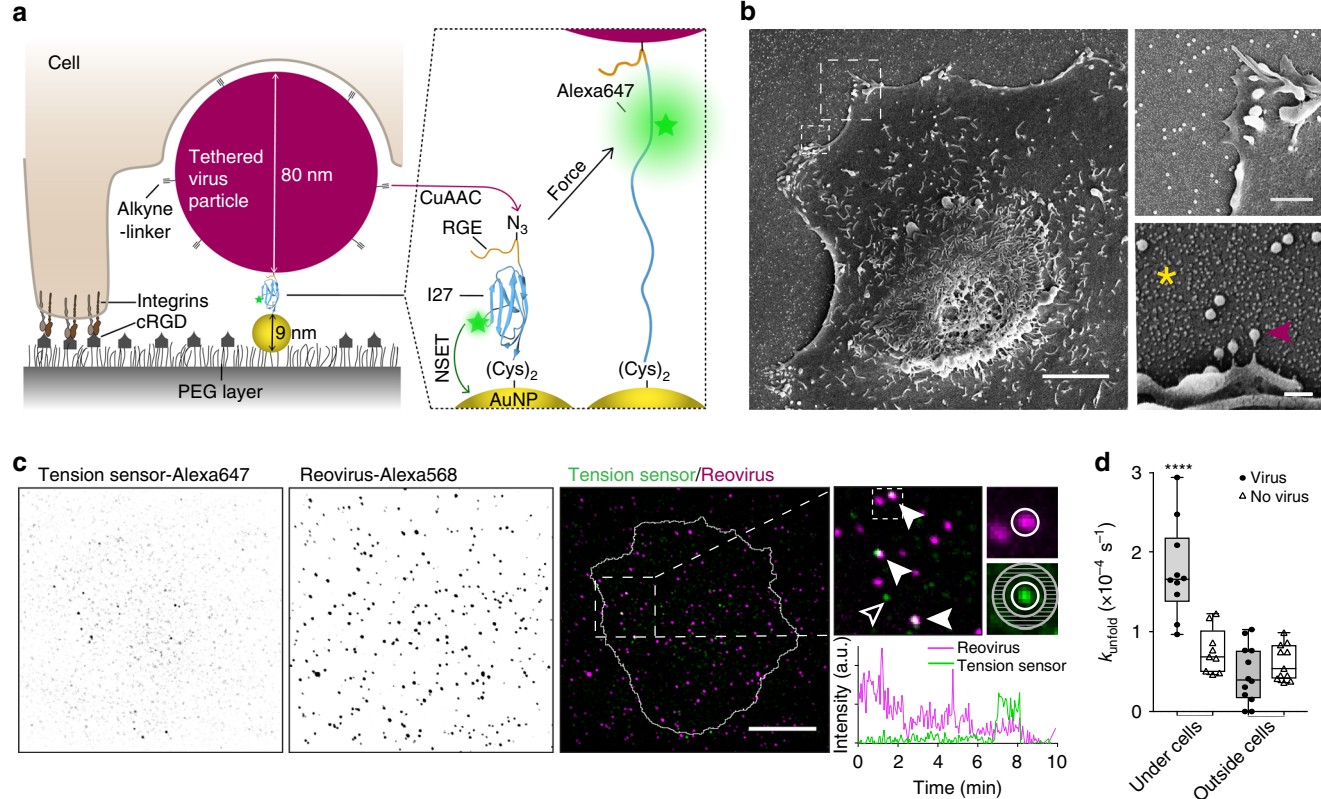

**Fig. 1 Molecular tension sensors detect forces on virus particles exerted by the ventral cell side. a** Schematic of a cell interacting with a tethered virus particle from its ventral side (not to scale). PEG-passivated glass coverslips were decorated with cRGD ligands (gray) for integrin-specific cell adhesion and tension sensors consisting of gold nanoparticles (AuNPs, yellow) and Alexa647-labeled titin I27 domain (blue). Alkyne-modified reoviruses (magenta) were tethered via click chemistry (CuAAC). Fluorescence of the tension probes is quenched by the AuNP via nanometal surface energy transfer (NSET) and increases upon unfolding. **b** SEM images of a BSC1 cell on an array of tension sensors (asterisk) onto which single virus particles are bound (arrowhead). Scale bars, 5 μm (overview), 1 μm (upper), 200 nm (lower panel). **c** Maximum projections of TIRF images over 10 min of titin-based tension sensors (green) and fluorescently labeled reoviruses (magenta) and merge with the cell outline (see Supplementary Movie 1). Arrowheads indicate signals of opened tension sensors at virus sites. Hollow arrowhead points towards a nonspecific tension signal. Scale bar, 10 μm. Fluorescence intensity traces were analyzed at sites of viruses (white circle) or at random spots not co-localizing with viruses after subtraction of the local background signal (gray annulus). Reovirus signal decays over time due to photobleaching of the sparsely Alexa568-labeled virus. **d** Rate $k_{unfold}$ of tension sensors with characteristic increase in fluorescence representing the tension sensor unfolding at sites of virus immobilization/at random spots under the cells and in a control region outside of cells during 10 min, 1 h post seeding cells ($n = 10, 11, 12, 13$ cells or control regions, with a total number of 114 events out of 1073 virus particles, 88 non-specific events out of 2549 random spots underneath the cells, 40 events per 1228 viruses and 80 non-specific events out of 2713 random spots in the control region, respectively, three technical replicates, box-plots represent median ± 95% CI with whiskers to the min and max values, ****$P < 0.0001$, one-way ANOVA with post hoc Tukey). Source data are provided as a source data file.

molecular unbinding[23]:

$$k_{unfold} = k_{unfold}^{0} \times e^{\frac{F\Delta x}{k_B T}} \quad (2)$$

where $\Delta x$ is the distance between the bound state and the energy barrier along the unfolding pathway, $k_B$ the Boltzmann's constant and $T$ the temperature. Since we cannot resolve the force history applied to single virus particles with our method, to simplify the estimation of mechanical interactions at the cell-particle interface, we assume here that cells exert a constant force $F$ on all particles upon contact. Combining and rearranging Eqs. (1) and (2) allows to infer the minimal mean force, which a 'mean particle', i.e., the average of an ensemble of particles, would experience at experimental conditions (37 C):

$$F = \ln\left(\frac{k_{unfold} - k_{unfold}^{0}}{k_{unfold}^{0}} + 1\right)\frac{k_B T}{\Delta x} = 44\,(\pm 3)pN \quad (3)$$

with the unstressed off-rate $k_{unfold}^{0} = 3.3 \times 10^{-4}\,s^{-1}$ and $\Delta x = 0.25$ nm from Carrion-Vazquez et al.[24] corrected for the imaging rate (100 ms illumination every 3 s). Even though titin

has been reported to withstand significantly higher forces up to 250 pN[25], such absolute values depend on pulling speed and thus loading rates, which we disregard in our simple estimate. Of note, the obtained value is in agreement with the previously observed receptor-mediated forces acting on the I27 tension probe at minimal loading rates[20].

We conclude that cells actively exert forces on the viruses that are presented in a cell-matrix context with ~44 pN being a reasonable lower estimate as we disregard loading rate effects. To limit the possibility that a virus particle binds to multiple tension probes on the surface, we lowered their concentration by 20-fold compared to experiments with monovalent RGD ligands (Supplementary Fig. 1). Indeed, signals occurring at sites of virus immobilization in the control region and background signals not associated with virus particles occurred at similar rates (Fig. 1d), suggesting that in both cases single tension probes were assessed. However, we cannot exclude multiple bonds, which would further dramatically increase the mechanical resistance. Nonetheless, the herein found minimal force falls into the range of interaction strengths of 10–58 pN between influenza[12,18], dengue[19], rhino[11],

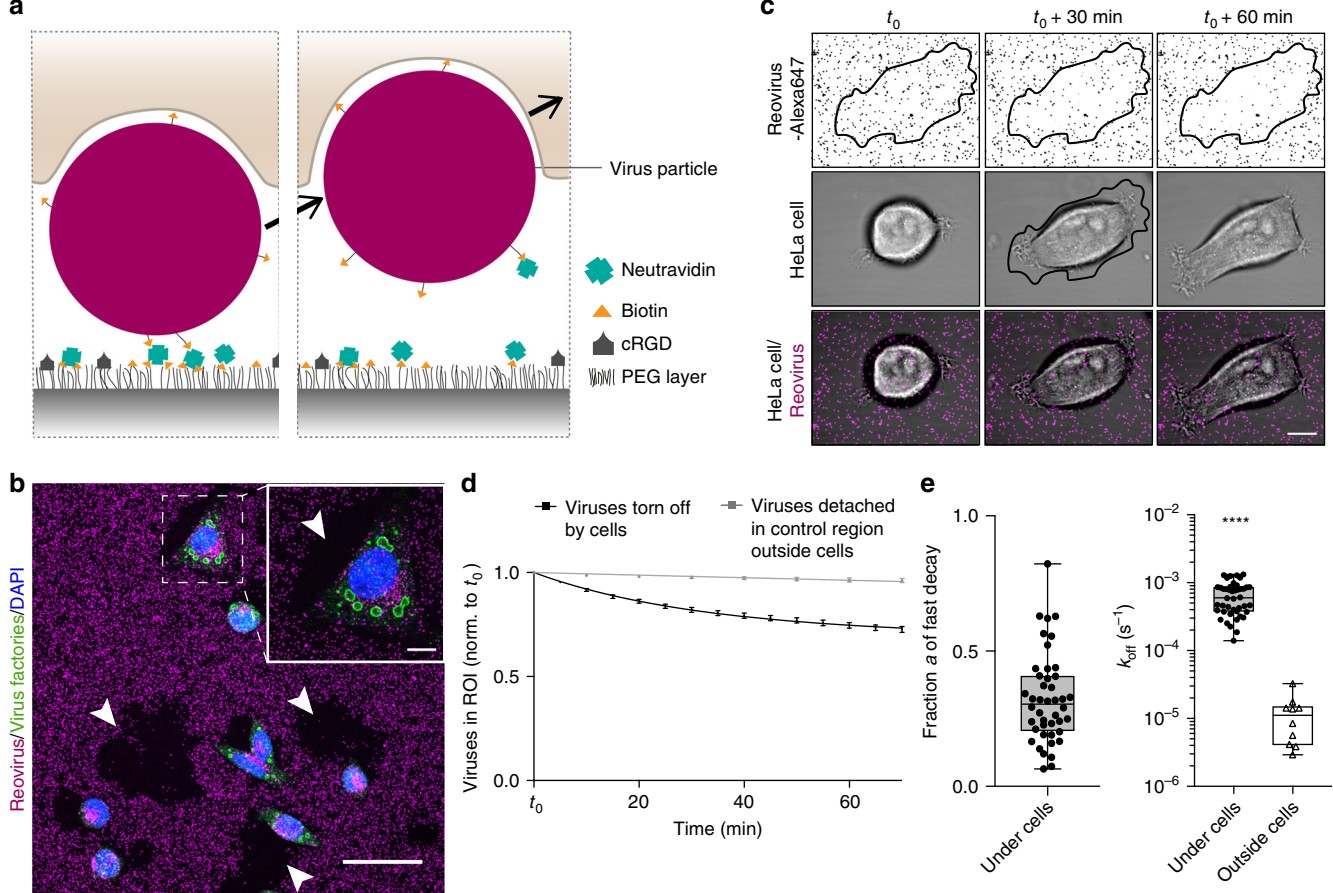

**Fig. 2 Forces during virus uptake from the ventral cell side exceed biotin-neutravidin unbinding force. a** Schematic of a cell tearing off a biotin-neutravidin surface-bound virus particle. **b** Immunostaining for viral infection in HeLa cells (green, virus factories) upon tearing off biotin-neutravidin bound reovirus particles (magenta) from the surface. Arrowheads point towards regions where cells removed viruses completely from the surface. Scale bars, 50 and 10 μm in zoom-in. **c** Confocal time-lapse images of a HeLa cell (transmission) spreading and tearing off Alexa647-labeled reovirus particles (magenta) from the biotin-neutravidin surface with the cell outline at $t_0 + 30$ min defining the region of interest (ROI). Scale bar, 10 μm. **d** Normalized number of virus particles over time either in the ROI under HeLa cells or in a control region outside cells. Data are represented as mean ± s.e.m. and fitted with two-phase decay or single exponential decay functions for ROI under cells and outside cells, respectively. **e** Fitting parameters $a$, the fraction, and $k_{off}$, the off rate of particles being actively removed by cells or detaching outside cells. ($n = 44$ cells with a total of 5598 viruses and 10 control regions with 6892 total viruses from seven technical replicates. Box-plots represent median ± 95% CI with whiskers to the min and max values, ****$P < 0.0001$ according to unpaired two-tailed Mann–Whitney test). Source data are provided as a source data file.

pseudotyped rabies viruses[13] or enterovirus 71[17] and the dorsal side of cells as observed with force spectroscopy methods by other investigators.

**Cells rupture viruses that are immobilized by biotin**. To investigate if the observed forces can induce particle uptake by the cells, we next immobilized reoviruses non-covalently via biotin-neutravidin on a PEGylated glass substrate co-presenting cRGD ligands to enable cell adhesion (Fig. 2a). Biotinylation of reovirus particles at a final labeling concentration of 8.4 μM biotin-NHS did not reduce their infectivity (Supplementary Fig. 3a, b). Further, we confirmed that biotinylated viruses specifically attached to neutravidin on the $SiO_2$-PEG-biotin surfaces (Supplementary Fig. 3c, d). Intriguingly, we found that cells could indeed dissociate the biotin-neutravidin bonds and thus tore off viruses from the surface and internalized them causing viral infection (Fig. 2b). This was not observed with covalently immobilized viruses on homogeneous $SiO_2$-PEG-azide surfaces (Supplementary Fig. 4). To exclude that the rupture of viruses might be caused by protease degradation of neutravidin, we investigated virus-tearing from the surfaces before and after metalloprotease

inhibition with GM6001 (25 μM) by confocal time-lapse microscopy. No significant difference in tearing of surface-bound viruses was observed and after 4 h most of the particles underneath the cell surface were removed by untreated as well as metalloprotease-inhibited cells (Supplementary Fig. 5a, b and Supplementary Movie 2).

We therefore conclude that force-driven unbinding occurs, also as previously shown for integrin-mediated forces[26], further corroborated by the finding that in the control region outside of cells, almost no dissociation of virus particles from the surface could be observed.

**Kinetic analysis of virus detachment from the surface**. To quantify these effects, we analyzed the kinetics of virus detachment from the biotin-neutravidin surfaces starting at 30 min post seeding cells ($t_0$) in a region of interest (ROI) defined as the projected cell area at $t_0 + 30$ min (Fig. 2c and Supplementary Movie 3, see Methods). We observed that many particles were displaced by the cells in x-y direction either before or after uptake. However, due to the limited z-resolution of the confocal images, it was not possible to determine whether these particles are below or

above the cell membrane. We hence only considered the removal of particles from the initial surface-bound position as the bond breaking event, independent of their exact z-position (Fig. 2d). The number of virus particles remaining bound to the surface over time followed first-order unbinding kinetics and we fitted an exponential decay function:

$$c_{\text{Particles}}(t) = c_{\text{Particles}}(t_0) \times e^{-k_{\text{off}} \cdot t} \tag{4}$$

with normalized number of virus particles to the initial number $c_{\text{Particles}}(t_0) = 1$ over a 70-min time-lapse interval. Viruses in the control region outside of cells (with zero-force) showed an apparent off-rate of $k_{\text{off}}^0 = 1.2 \ (\pm 0.3) \times 10^{-5} \ \text{s}^{-1}$ (Fig. 2e). This is in the order of zero-force off-rates observed in comparison studies in literature for individually bound avidin[27] or streptavidin[28], whereas, individually bound neutravidin[29] has been found to dissociate two orders of magnitude faster likely due to the influence of deglycosylation. Here, multivalence of neutravidin combined with high biotin densities on the surface allows for multiple connections per virus particle, which can significantly increase their immobilization stability. Accordingly, these data suggest that multiple bonds per virus predominated in the experimental setup.

Cell induced virus detachment could not be fit with a simple exponential decay (which is discussed in the next paragraph) and we thus fitted two-phase decay functions:

$$c_{\text{Particels}}(t) = a \times e^{-k_{\text{off}} \cdot t} + (1 - a) \times e^{-k_{\text{off}}^0 \cdot t} \tag{5}$$

with $a$ representing the fraction of particles being actively torn-off during cell spreading, while the other fraction of particles $(1-a)$ dissociates with background kinetics. We found that HeLa cells actively tear-off a fraction $a = 32 \ (\pm 17)\%$ of surface-immobilized reoviruses during spreading (Fig. 2e). For these viruses, we then find the mean off-rate $k_{\text{off}}^{\text{HeLa}} = 6.5 \ (\pm 0.5) \times 10^{-4} \ \text{s}^{-1}$. Applying again Bell's model (Eq. 2) with the distance to the biotin-neutravidin transition state $\Delta x = 0.5 \ \text{nm}$[30], this translates into a mean force of 34.4 $(\pm 2.2)$ pN per particle.

Interestingly, U373 cells, which spread out to larger areas, were found to tear-off more viruses $(a = 48 \ (\pm 21)\%)$, however, the kinetics and thus forces exerted on viruses $(k_{\text{off}}^{\text{U373}} = 4.5 \ (\pm 1.3) \times 10^{-4} \ \text{s}^{-1}$ and $F^{\text{U373}} = 31.3 \ (\pm 3.2)$ pN) were similar those of HeLa cells (Supplementary Fig. 5c–e).

Noteworthy, the load on individual particles might be even higher than the one inferred from the approximation of a mean particle, since the loading rates and force history can vary, allowing to break molecular bonds with much lower force[26]. Hence, the mean forces are in agreement with the results obtained from the molecular tension probes (Fig. 1).

**Particles are preferentially removed at the cell periphery**. We observed that virus removal is enhanced during initial attachment at the cell periphery, as the virus detachment plateaus when cells spread above the ROI (Fig. 2d). This suggests a facilitated uptake at the lamellipodium, where close contacts to the substrate are established via integrins[31]. Indeed, simultaneous imaging of the relative distance between cells and the substrate and virus-sized nanoparticles (which are discussed in detail hereafter) by inter-ference reflection microscopy showed almost no uptake during early adhesion and in the loosely adhering center of the cell (Fig. 3a and Supplementary Movie 4). To quantify this spatial preference and account for the cell spreading we traced the time $t^*$ each particle stayed underneath the projected cell area before removal from the surface considering multiple ROIs (Fig. 3b). Particles with short interaction times with the ventral cell side, namely at the cell edge, are removed faster as the decay plateaus for longer interaction times (Fig. 3c). This confirms that cells

preferentially tear off particles underneath the cell edge. We fitted again a two-phase decay function (Eq. 5) and compared the fit parameters for both analysis approaches (Fig. 3d). Since the results did not significantly differ between the analysis using multiple ROI and single ROI, the latter one was thereafter chosen for the sake of simplicity.

**Design of particles mimicking virus size and capsid motifs**. We next addressed the role of the adhesion energy between viruses and cells. Reoviruses, like many viruses, interact with cell surface receptors, such as JAM-A and $\beta_1$ integrins, which mediates their internalization[9]. The resulting gain in adhesion energy between particles and the plasma membrane might be sufficient to drive particle wrapping and to modulate cellular uptake of nanoscale objects[32,33].

To test the influence of cargo properties on the uptake mechanics, we employed 100 nm AuNPs mimicking virus size and integrin-binding motifs at the surface of viruses. AuNPs were decorated with an inert PEG layer to diminish non-specific adsorption of serum proteins[34]. Specific functionalization with biotin linkers, a fluorescent dye (StarRed) and cell adhesion ligands (cRGD, which has a high affinity for integrins $\alpha_v\beta_3$ or integrin $\alpha_5\beta_1$ selective ligand[35]) was achieved using click chemistry on modified PEG-linkers (Fig. 4a and Supplementary Fig. 6a, b). We characterized the binding of these ligands to the nanoparticles by dynamic light scattering and fluorescent binding assay (see the "Methods" section). The hydrodynamic diameter of functionalized particles was 127 $(\pm 3)$ nm, irrespective of the presence of cell adhesion ligands (Supplementary Fig. 6c). Considering that the cell-substrate distance is ~80 nm at lamellipodia, as recently reported in ref. [31], both virus and nanoparticles would strongly push against the ventral side of the cell simply due to confinement. Successful functionalization of AuNPs was confirmed by a change in zeta potential from −15 $(\pm 5)$ mV to −21 $(\pm 3)$ mV and −10 $(\pm 1)$ mV for AuNPs without adhesion-promoting ligands and cRGD or integrin $\alpha_5\beta_1$ selective ligands, respectively (Supplementary Fig. 6d). The ligand density on AuNPs after click reaction was $10^{18}$ ligands per m², which allows unconstrained receptor binding (see Supplementary Fig. 6e–g and note 1).

**Removal of surface-bound particles depends on adhesion**. To analyze the uptake kinetics, we immobilized the biotinylated AuNPs similarly to viruses via neutravidin on matrix-mimetic substrates (Fig. 4a) and checked for their monodispersity by SEM (Supplementary Fig. 6h). We likewise monitored particle removal from the substrate by confocal microscopy over 70 min and fitted an exponential decay function to normalized numbers of surface-bound AuNPs, which revealed the apparent dissocia-tion rate from the surface in absence of cells of $k_{\text{off}}^{0,\text{AuNPs}} = 1.2 \ (\pm 0.1) \times 10^{-5} \ \text{s}^{-1}$.

Interestingly, HeLa cells were capable of tearing off nanopar-ticles even without specific ligands (Fig. 4b, c and Supplementary Movie 5). However, they did so at a lower rate $k_{\text{off}}^{\text{AuNPs}(1)} = 3.2 \ (\pm 0.8) \times 10^{-4} \ \text{s}^{-1}$ and thus the exerted forces of 28.0 $(\pm 2.0)$ pN were lower than the forces on biotinylated reoviruses. When cells were able to specifically interact via integrins with particles bearing cRGD ligands, the tearing off biotin-neutravidin immobilized AuNPs was significantly enhanced (Fig. 4c and Supplementary Movie 6). This effect proved to be independent of cell-matrix interactions, as spreading on the matrix-mimetic surfaces was not influenced by the presentation of additional cell adhesion ligands on the AuNPs (Fig. 4d). Of note, the surface density of cRGD immobilized on the surface background is much higher than total adhesion sites provided by the AuNPs.

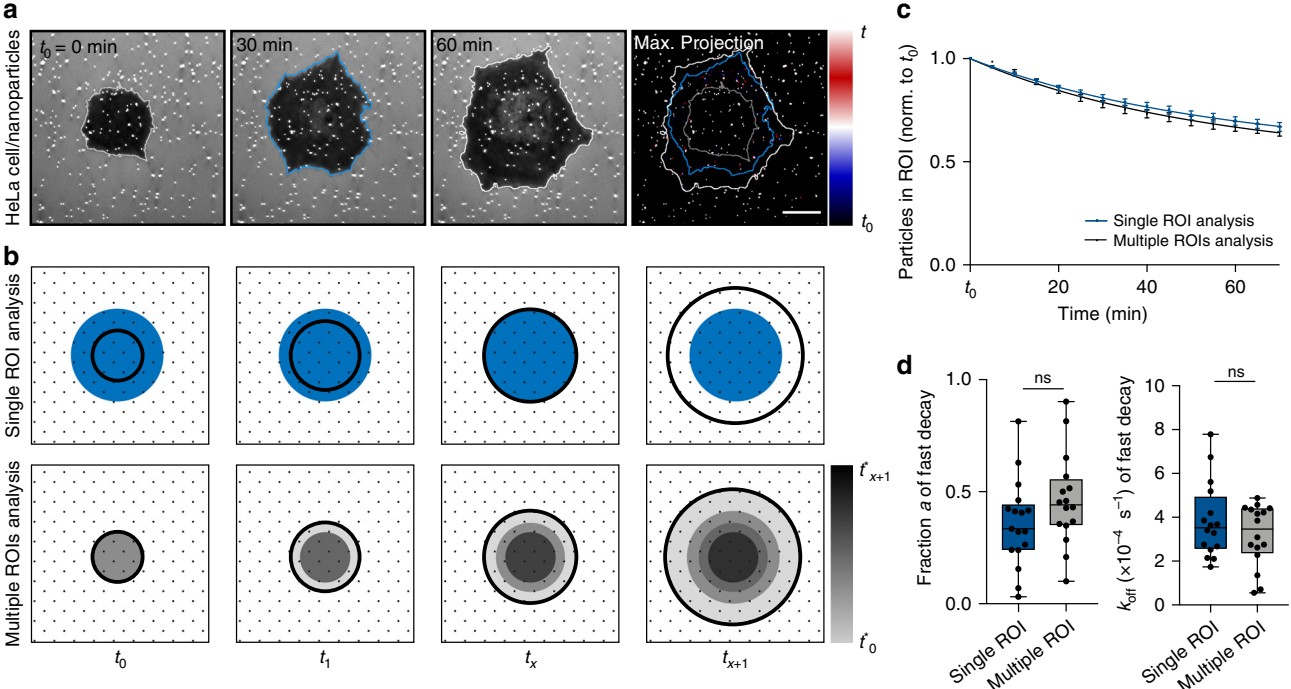

**Fig. 3 Kinetic analysis of removal of surface-bound particles shows spatial preference for cell periphery. a** Interference reflection microscopy images of a HeLa cell spreading and tearing off biotin-neutravidin surface immobilized virus-sized gold nanoparticles. Where the cell formed close contacts with the substrate, light reflection and interference with the incident light creates the dark contrast. The nanoparticles scattered the incident light and thus appear bright. In the maximum projection with temporal color-code stable particles are displayed in white, while particles that were removed within the first 30 min are blue and particles that were moved between 30 and 60 min appear red and the cell outline at $t_0 + 30$ min is printed in blue. Scale bar, 10 µm. **b** Schematic of the regions of interest (ROIs) for single ROI analysis and analysis using multiple ROIs. In the single ROI analysis, we considered particles, which fall within the blue ROI defined as the projected cell area (black circles) at $t_x = t_0 + 30$ min Data were normalized to the particles in this ROI at $t_0$. In the analysis using multiple ROIs we grouped particles (similar gray areas), that stayed underneath the projected cell area for at least the indicated interaction time $t^*$. Data were normalized to the available number of particles per interaction time and corrected for the already torn off particles at all shorter times. **c** Normalized number of surface-bound nanoparticles without ligands under HeLa cells over time either analyzed in a single ROI or multiple ROIs. Data are represented as mean ± s.e.m. and fitted with two-phase decay functions. **d** For the particles being actively removed by the HeLa cells, the fraction $a$ and the off rate $k_{off}$ are shown as obtained by fitting from the two-phase-decay of particles ($n = 16$ cells, 3 technical replicates, box-plots represent median ± 95% CI with whiskers to the min and max values, one-way ANOVA with post hoc Tukey test). Source data are provided as a source data file.

Interestingly, the specific adhesion between nanoparticles and cells influences particle uptake at two levels: first, the fraction $a$ of particles being torn off during cell spreading increased significantly (Fig. 4e). We interpret this as an enhanced probability to establish contact between the membrane and the particle, which starts the uptake process and can be compared to the nucleation step in processes of nucleation and growth[36]. Second, the tear off rate was not significantly faster ($k_{off}^{AuNPs(2)} = 3.9 \,(\pm0.5) \times 10^{-4}\,s^{-1}$), indicating similar interaction forces of 29.7 (±1.1) pN.

While HeLa cells express several RGD-binding integrin subunits such as $\alpha_3$, $\alpha_5$ and $\beta_1$[9] only low surface expression levels of the high affinity receptor $\alpha_v\beta_3$ were reported by Maginnis et al.[9]. To test for stronger receptor-ligand interactions, we investigated how selective ligands for $\alpha_5\beta_1$ integrins affect the particle uptake in HeLa cells. Therefore, we coated the AuNPs with a peptidomimetic integrin ligand, which has ~1000 fold higher affinity for $\alpha_5\beta_1$ over $\alpha_v\beta_3$[35,37]. Remarkably, almost all AuNPs with $\alpha_5\beta_1$-specific ligands were torn-off during cell spreading at a rate more than twice as fast with $k_{off}^{AuNPs(3)} = 9.4 \,(\pm0.3) \times 10^{-4}\,s^{-1}$ than AuNPs with cRGD (Fig. 4d, e and Supplementary Movie 7), indicating forces as high as 37.2 (±0.3) pN. Receptor blocking with an antibody against $\beta_1$ integrin, could partially reduce the apparent dissociation rate. Uptake of AuNPs with cRGD remained unchanged presumably

due to interactions with other integrin types (Supplementary Fig. 7). Note that cell spreading was reduced upon blocking of $\beta_1$ integrins and all data were corrected for the respective projected cell area 60 min post seeding during the analysis (Fig. 3). However, the observed change in the tear off kinetics was specific for AuNPs with $\alpha_5\beta_1$-specific ligands and hence uncoupled from the reduced cell area.

We conclude that enhanced adhesion between nanoparticles and cells increases the probability of uptake. Further, we could show that the forces during surface-removal of particles can be tuned via specific interactions, increasing the dissociation rate from matrix-mimetic surfaces. However, even non-specific interacting particles were torn-off by the ventral cell side, albeit with lower probability. In contrast to earlier reports on the uptake of passivated nanoparticles from solution[38], our experimental setup allows to characterize the influence of particle adhesion decoupled from the likelihood of a specific encounter due to the spatial confinement of particles in close vicinity to the cells.

**Actomyosin inhibition reduces particle rupture from surface.** To test if the uptake forces were generated by actin or myosin we treated the cells with the actin polymerization inhibitors Cytochalasin D and LatrunculinA or ROCK inhibitor Y-27632. Since these drugs strongly impair cellular attachment and spreading, we

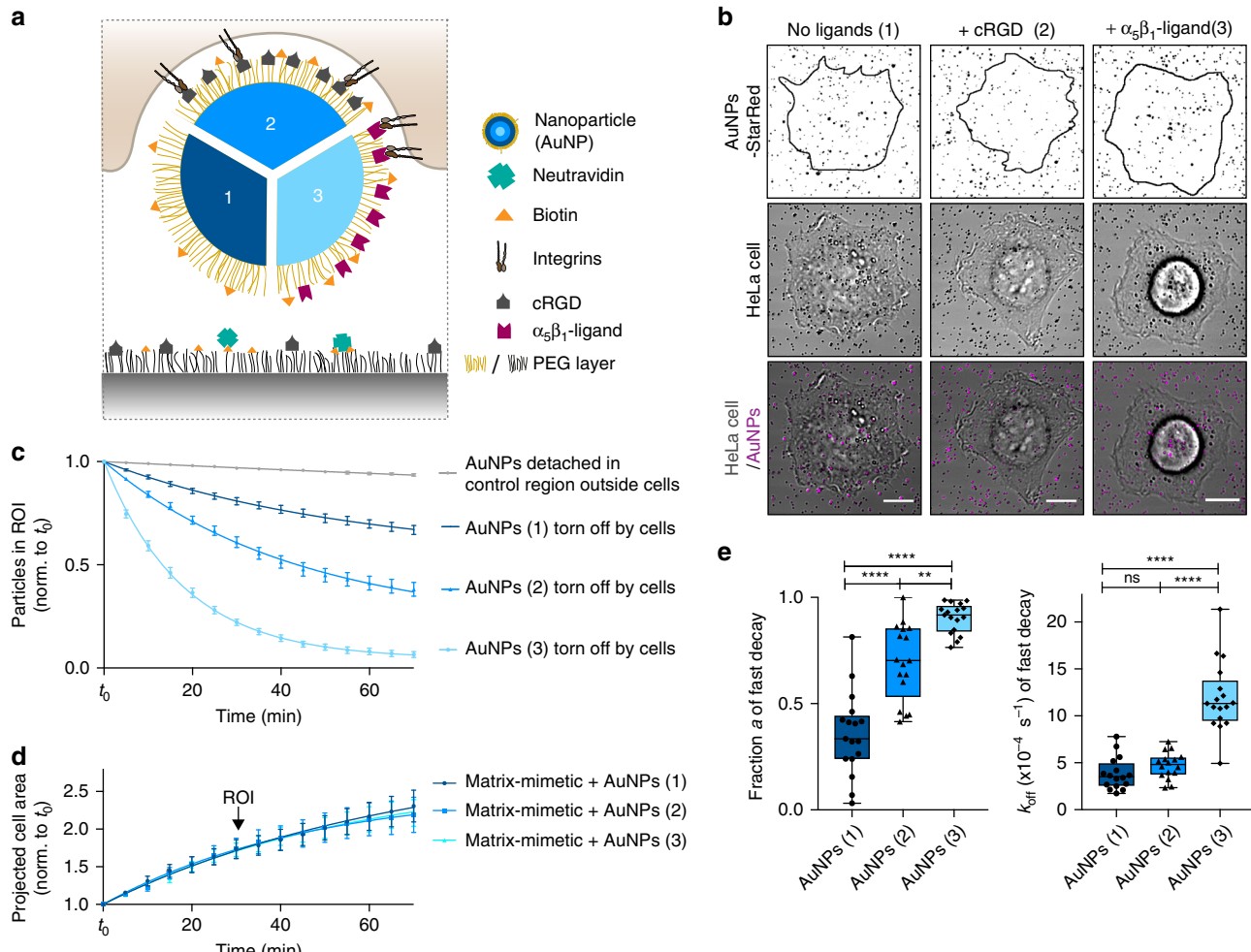

**Fig. 4 Receptor-ligand interactions enhance nanoparticle uptake from the ventral cell side. a** Schematic of a cell tearing off a biotin-neutravidin bound nanoparticle (AuNP) either only passivated with PEG (1) or additionally functionalized with cRGD (2) or integrin $\alpha_5\beta_1$ selective ligands (3). **b** Confocal images of a HeLa cell (transmission) spreading and tearing off StarRed-fluorescently labeled AuNPs (magenta) from the surface at $t = t_0 + 70$ min with ROIs 1 h post seeding. Scale bars, 10 μm. **c** Relative number of particles in the ROI under the cells or in a control region outside the cells over time. Data are represented as mean ± s.e.m. and fitted with two-phase decay functions. ($n = 3$ control regions or 19, 24, 27 cells on AuNPs (1), (2) and (3), respectively, three technical replicates, data are represented as mean ± s.e.m. and fitted with an exponential decay function). **d** Projected cell area normalized to $t_0$ of cells spreading on the matrix-mimetic surfaces decorated with AuNPs 1–3 over time with exponential fits. **e** For the particles being actively removed by the HeLa cells, the fraction $a$ and the off rate $k_{off}$ are shown as obtained by fitting from the two-phase-decay of particles in the ROI as presented in (**c**). (Box-plots represent median ± 95% CI with whiskers to the min and max values, $**P < 0.001$, $****P < 0.0001$; one-way ANOVA with post hoc Tukey). Source data are provided as a source data file.

seeded the cells on the matrix-mimetic surfaces for 60 min before treatment. To determine the effective concentrations in HeLa cells, we observed the integrity of actin filaments by immunofluorescence staining 90 min after adding the inhibitors (Fig. 5a). Sixty nanometers of Cytochalasin D, 100 nM Latrunculin A and 20 μM Y-27632 final concentrations were chosen since these disrupted the cortical actin and the organization of the actin cytoskeleton, respectively, while still allowing the cells to adhere to the substrate. As expected cell spreading was impaired upon blocking of actin polymerization whereas increased cell spreading followed inhibition of ROCK-mediated cellular contraction (Fig. 5b) as previously reported for epithelial cells[39,40]. To test the effect on the tearing of biotin-neutravidin immobilized nanoparticles, we normalized the number of particles in the ROI underneath the cells to the first timepoint after addition of the drug (Fig. 5c). Note that due to the plateauing of the exponential decays these fits cannot be directly compared to the previous data but only among each other. We observe that inhibition of actin

polymerization significantly reduced particle tearing off the surface, while it was only slightly reduced by ROCK inhibition. The two-phase exponential fits show that this effect was mainly due to a lower fraction $a$ of particles being actively moved, while for these particles the rate of decay remained similar or was even slightly increased by actin/myosin interference (Fig. 5d). This suggest that actin/myosin is important to maintain close surface adhesion but plays a minor role for the generation of the uptake forces on nanoparticles. However, these results have to be taken with care since the uptake of particles from the surface depends on the spreading process, which differed between the conditions and cellular fitness was generally impaired (Supplementary Movie 8–10).

**Adhesion energy density estimated by mathematical modeling.** Hence, we hypothesize, that forces generated by the adhesion energy between particles and cells are enough to counteract the

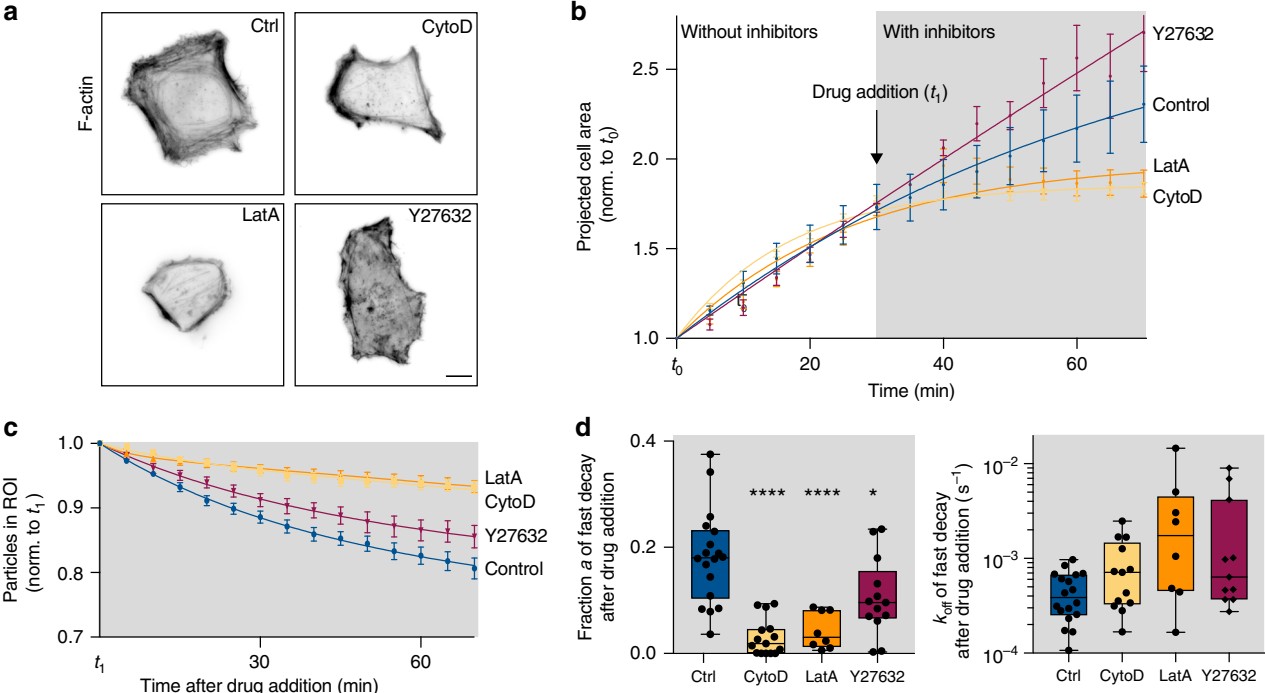

**Fig. 5 Actin/myosin inhibition significantly reduces tearing off surface-bound particles. a** Inverted wide-field fluorescence images of HeLa cells fixed 90 min post drug addition. Actin filaments were immunolabeled with phalloidin-TRITC. Scale bar = 10 μm. **b** Projected cell area normalized to $t_0$ of cells spreading on the matrix-mimetic surfaces decorated with nonspecific AuNPs (#1, see Fig. 4) over time. Actomyosin disrupting drugs Cytochalasin D (CytoD, 60 nM), Latrunculin A (LatA, 100 nM) or ROCK inhibitor (Y27632, 20 μM) were added 1 h after seeding of the cells. **c** Relative number of particles in the ROI under the cells over time after drug treatment. ($n$ = 19, 15, 14, 15 cells, 2 technical repeats, data are represented as mean ± s.e.m. and fitted with an exponential decay function) **d** For the particles being actively removed by the HeLa cells, the fraction $a$ and the off rate $k_{off}$ are shown as obtained by fitting from the two-phase-decay of particles in the ROI. (Box-plots represent median ± 95% CI with whiskers to the min and max values, *$P$ = 0.0145, ****$P$ < 0.0001; one-way ANOVA with post hoc Tukey). Source data are provided as a source data file.

energetic cost for membrane bending during particle uptake, when membrane deformation arising from particle confinement initiates uptake.

To investigate the adhesion energy provided by reovirus and nanoparticles in our experiments, we modeled the energetic cost of particle uptake as a function of bending rigidity and surface tension of the membrane[41]. Reovirus or nanoparticles are considered as spheres with radius $R$, which can be covered with homogeneously distributed ligands and adhere to membrane surface receptors along the area (Fig. 6a). In our model wrapping is driven by the energetic gain of particle adhesion to the cell membrane, either specifically to receptors or non-specifically. In contrast, uptake is counteracted by the energetic cost for bending the membrane and increasing the membrane area[42]. We only consider the deformation energy of the membrane that is bound to the particle, which is restricted to the shape of a spherical cap with $A_{ad} = 2\pi R^2(1 - \cos\theta)$. The progress of particle uptake is described by the uptake angle $\theta$. As the free membrane is ~80 nm above the substrate[34] we assume that initially the particles are already halfway wrapped. The Helfrich-Hamiltonian, giving the total free energy of particle uptake, reads[43,44]

$$E_{total} = -E_W + E_\kappa + E_\sigma = -WA_{ad} + \frac{1}{2}\left(2\kappa H^2 A_{ad} + \sigma \Delta A_{ad}\right)$$

(6)

with $E_W$ representing the adhesion energy gain, where $W$ is the energy density of the receptor populated nanoparticle or virus surface times the adhesion area $A_{ad}$. $E_\kappa$ is representing the membrane bending energy where $\kappa$ is the bending rigidity and $H = \frac{1}{R}$ the mean membrane curvature and $E_\sigma$ is representing the energetic cost for increasing the membrane area, where $\sigma$ is the

membrane tension and $\Delta A_{ad} = \pi R^2(1 - \cos\theta)^2$ the increase in adhesion area. As we assume that the particle is initially already halfway wrapped, the energy counteracting uptake is reduced by a factor of 2. The different energy contributions are shown as a function of $\theta$ normalized to the maximum of the adhesion energy (Fig. 6b). As the particle adheres to the cell membrane it experiences a force in z-direction[45]. This thermodynamic uptake force can be calculated by taking the derivative of the total energy with respect to z, $F_z = -\frac{dE_{total}}{dz}$. Then the adhesion energy can be calculated as a function of the uptake force

$$W(F_z) = c_0 + c_1 F_z$$

(7)

with the constants $c_0 = \frac{\kappa}{R^2} + \frac{\sigma z}{2R}$ and $c_1 = \frac{1}{2\pi R}$. Thus we find a linear relation between uptake force and adhesion energy considering a mean bending rigidity $\kappa = 25$ $k_B$T[44,46], surface tension[47] $\sigma = 10^{-5}$ Nm$^{-1}$ and an estimated rupture length $z = 10$ nm (solid line in Fig. 6c). The adhesion energy density for AuNPs without specific ligands was in the order of $W^{AuNPs(1)}$ $^{(2)} = 0.10$ mJ m$^{-2}$, while integrin $\alpha_5\beta_1$ selective ligands increased the adhesion energy to $W^{AuNPs(3)} = 0.12$ mJ m$^{-2}$ (symbols in Fig. 6c). By using the additional adhesion energy $\Delta W = 0.02$ mJ m$^{-2}$, we can estimate the number of nanoparticle ligands that bind to cell surface receptors. With the typical energy contribution of $\Delta\epsilon = 10$ $k_B$T per receptor-ligand bond[42] and distributing this energy homogeneously over the particle surface we found $N = (\Delta W/\Delta\epsilon)\cdot 4\pi R^2 = 25$ additional ligands bind to enhance the uptake. The high non-specific component could arise from serum proteins and other biomolecules adsorbing on the nanoparticles despite passivation[48]. While increased passivation, e.g., with varying PEG length or without biotin ligands and

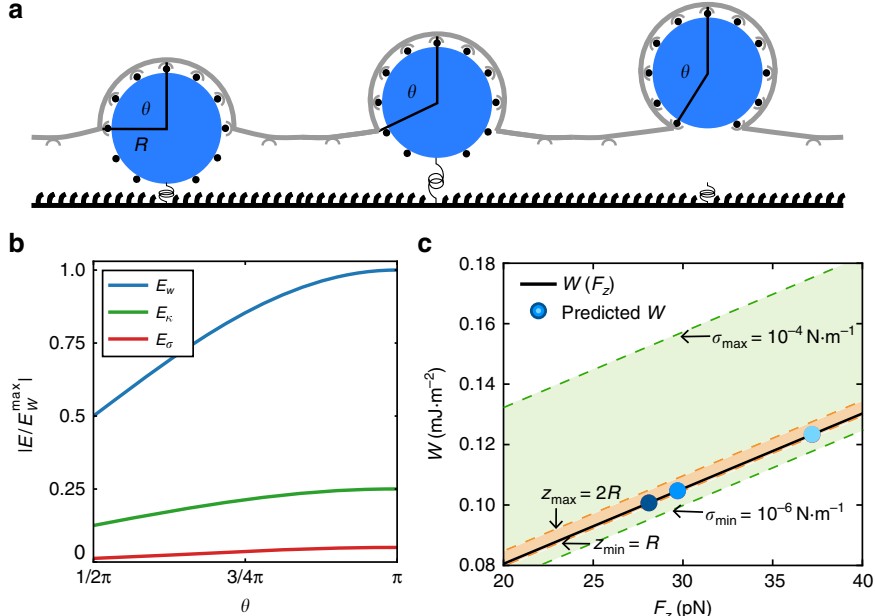

**Fig. 6 Mathematical model of nanoparticle uptake provides calculated adhesion energy. a** In the model the particle (blue) with radius $R$, immobilized by a force sensor (spring) is homogenously covered with ligands (black dots) that can adhere to the cell surface receptors (gray). We assume that initially the particle is already halfway wrapped. **b** Normalized energy values of contributions from adhesion energy, $E_W$ (blue), curvature energy, $E_\kappa$ (green), and stress energy, $E_\sigma$ (red), calculated according to Eq. 6. **c** Calculated adhesion energy density $W$ as a function of uptake force (solid line). The orange and green region shows the uncertainty of the adhesion energy caused by the uncertainty of the rupture length $z$ and surface tension $\sigma$, respectively. The blue points show the predicted adhesion energy for the uptake forces of AuNPs with three different coatings.

fluorophores could possibly limit this effect, biologically relevant particles such as viruses inherently show a non-specific binding, e.g., to glycoproteins and could likewise interact with extracellular biomolecules accounting for the non-specific adhesion component.

For reovirus, which is smaller than the decorated nanoparticles applied here ($R = 45$ nm as measured by DLS in ref. [21]), a higher adhesion energy density of $W^{\mathrm{reovirus}} = 0.18$ mJ m$^{-2}$ is expected to overcome the membrane bending. With the non-specific adhesion energy, we found from the nanoparticle uptake, the additional adhesion energy is $\Delta W = 0.08$ mJ m$^{-2}$. This corresponds to a mean number of 30 specific ligand-receptor interactions, which could be provided by the reovirus capsid proteins $\sigma 1$ and $\lambda_2$ (see Supplementary note 2).

In this model we use the average force per particle to describe a mean bond, which neglects that different virus-cell interactions could contribute different energies. Further, these numbers strongly depend on the surface tension and thus the mechanical state of the cell. Hence by varying the extracellular environment, particle uptake can be triggered towards a regime with selective particle uptake depending on the adhesion energy[2]. Here we could show that the adhesion energy between particles and cells mediates their uptake, but it cannot be ruled out that further energy contribution, e.g., from the endocytic machinery also play a role.

## Discussion

In summary, we introduce two methods allowing to infer forces exerted on viruses at the ventral cell side, namely the use of tension sensors and the quantitative analysis of uptake kinetics. Here we employed reovirus, which is an established model for receptor-mediated endocytosis, however, both techniques can be easily adopted to other non-enveloped viruses. Results from viral capsid-mimicking nanoparticles demonstrate that mechanical forces mediating viral uptake are a general phenomenon that

transcends the virus itself and they can be tuned by specific interactions. Differences in the uptake of nanoparticles coated with cRGD, mainly targeting integrin $\alpha_v\beta_3$, versus particles with $\alpha_5\beta_1$ selective ligands could arise from the different roles of the integrin subtypes. While integrin $\beta_3$ was found to induce endocytosis in the absence of force[45], $\alpha_5\beta_1$ determines adhesion strength[49] and mediates reovirus entry[9]. The emerging mechanical aspect of specific reovirus-cell interactions was previously overlooked even though our model suggests that the adhesion energy is sufficient to account for the observed forces.

Single molecule force sensors revealed forces acting on reovirus particles in the order of tens of pN as observed with TIRF live-cell imaging. To enable the detection of forces exerted on single viruses, we chose a low surface density of virus particles combined with a low concentration of tension probes. This improved the signal-to-noise ratio while still allowing for multiple bonds, which could not be quantified for the probes used in the current study. Another important aspect concerning the present set-up is the direction of force. Virus uptake induces a force in z-direction, however, shear forces dragging the particles in x-y along the plasma membrane prior to virus uptake will also lead to an unfolding of the titin I27-tension probe. It would be therefore of interest to map the orientation of the forces, e.g., by fluorescence polarization microscopy[50] and track the particles with higher temporal resolution.

Further, we observed the uptake of non-covalently surface-bound reoviruses and nanoparticles and analyzed the underlying forces by a kinetic model of molecular bond rupture. We found that an enhanced interaction strength between the particles and the cells leads to higher probability of uptake, and higher forces. However, even particles without specific ligands were torn-off by the ventral side of cells. Hence, we speculate that the close contact to the cell and non-specific adhesion are enough to drive particle wrapping. This is consistent with the enhanced uptake of particles at sites of close contact with the substrate and recent reports[21,51]

showing that nano-topographies on substrates lead to membrane curvature, which is sufficient to induce clathrin mediated endocytosis. Perturbation of actin polymerization and myosin contractility leads to reduced particle uptake, but our fitting procedures suggest that this effect is indirect, because it does not affect much the uptake kinetics itself. We conclude that particle confinement and adhesion energy might be sufficient factors to directly affect uptake mechanics.

Quantitative analysis of the uptake of non-covalently immobilized particles provides a general approach to study the impact of physicochemical properties of nanoparticles on their uptake. Furthermore, the influence of cellular properties like spreading, polarization[52] or the mechanical state[2] can be investigated with this setup in combination with small molecule inhibitors, micropatterns or soft substrates. Eventually a more profound knowledge on particle-cell interactions in a matrix environment can help to prevent entry of pathogens into host cells and is furthermore significant for the development and assessing the risks of nanoparticulate drug delivery tools[53].

## Methods

**Virus purification.** Mammalian reovirus strain T3D (originally obtained from Bernard N. Fields, Harvard Medical School) was grown in L-cells. Virus particles were purified according to standard protocols[54–56]. Briefly, infected cells were disrupted by sonication (UP200Ht ultrasonic processor, 50% of 200 W, 26 kHz, Hielscher Ultrasonics, Germany) for 40 s and lysed with 1% (v/v) of sodium desoxycholate (10% w/v) on ice for 15 min Virus particles were extracted by 1,1,2-trichlorotrifluoroethane (33% v/v), purified by ultracentrifugation on a CsCl gradient (1.25–1.45 g/cc) and dialyzed against virus buffer (150 mM NaCl, 10 mM MgCl₂, 10 mM tris(hydroxylmethyl) aminomethane (TRIS), pH 7.5) with 3 ml Slide-A-Lyzer dialysis cassettes (20000 MWCO, Thermo Fischer Scientific, USA) at 4 °C for 24 h. The concentration of reovirus particles in solution was determined to be around $1.7 \times 10^{11}$ particles per ml by nanoparticle tracking analysis with the NanoSight 300 (Malvern, UK).

**Surface modification of viruses.** Reovirus particles were randomly labeled at the lysine side chains on the viral capsid[57]. Therefore, reoviruses were first dialyzed against PBS to remove free amines in Slide-A-Lyzer MINI dialysis Devices, 3500 MWCO (Thermo Fischer Scientific, USA) at 4 °C overnight. Then 100 μl of the virus solution were mixed with the desired NHS-ligand and incubated at RT for 1 h. For experiments with covalently immobilized reovirus on single tension probes, propargyl-NHS at 33.2 μM final concentration and Alexa568-NHS (Thermo Fisher Scientific, USA) at 0.5 μM final concentration (for TIRF imaging) or Alexa647-NHS (Thermo Fisher Scientific, USA) at 32 μM final concentration (for fluorescence widefield and confocal imaging), and for experiments with non-covalently immobilized reovirus on neutravidin surfaces, 8.4 μM EZ-link sulfo-NHS-biotin and Alexa647-NHS at 33.7 μM final concentration (both Thermo Fisher Scientific, USA) were used. Particles were purified by gel filtration with Zeba spin columns (7000 MWCO, Thermo Fisher Scientific, USA).

**Labeling of SH-PEG-NH₂.** Twenty microliters of an 0.8 mM solution of SH-PEG-NH₂ (MW ~3000 g mol⁻¹, Iris Biotech, Germany) and 18 μl of 8 mM StarRed-NHS (Abberior, Germany), both in DMSO, were mixed and supplemented with 2 μl triethylamine. The reaction was allowed to proceed at RT for two days under continuous stirring and exclusion of light. No purification steps were necessary, since unbound dye molecules cannot specifically bind to the AuNPs and will be hence washed off after the coupling to gold. The product was stored at −20 °C until further use and will be referred to as SH-PEG-StarRed.

**Surface modification of AuNPs.** Gold nanoparticles (AuNPs, 100 nm diameter, OD1, in citrate buffer, Sigma Aldrich) were coated with a passivating PEG layer and functionalized as follows: 3 ml of the AuNP solution were incubated with 25 μM SH-PEG-alkyne (MW ~3000 g mol⁻¹, Biochempeg, USA) and 2.5 μM SH-PEG-StarRed at RT overnight under constant shaking and exclusion of light. Subsequently, unbound thiol linkers were washed off by centrifugation (three times, 2600 rcf at 4 °C for 10 min). After the third washing step the pellet was resuspended in 50 μl MilliQ water and the concentration was determined by UV–Vis absorption (NanoDrop ND-1000, Peqlab, Germany) and adjusted to $1 \times 10^{11}$ particles per ml. For non-covalent immobilization via biotin-neutavidin AuNPs were further functionalized with azide-EG₃-biotin (Jena Bioscience, Germany). Click reaction[58] between alkyne functions on the AuNPs with $5 \times 10^{10}$ particles per ml and the azide-EG₃-biotin (15 μM final concentration) was catalyzed by 100 μM CuSO₄ mixed with 500 μM tris(3-hydroxypropyltriazolylmethyl) amine (THPTA), 5 mM aminoguanidine and 5 mM sodium ascorbate in 100 mM phosphate buffer, pH 7. Optionally cRGD-azide (cyclo[RGDfE]K(N3), Peptide

Specialty Laboratories, Germany) (Supplementary Fig. 6a), integrin $\alpha_5\beta_1$-selective ligands[37] with azide modification (kindly provided by Prof. Dr. Horst Kessler, TU Munich, Germany) (Supplementary Fig. 6b) or azide-Fluor 488 (Jena Biosciences, Germany) were clicked in the same reaction at 300 μM final concentration. Modified AuNPs were purified from unbound ligands by size exclusion columns (Zeba Spin Desalting Columns, 7000 MWCO) and stored at 4 °C until further use for up to 4 days.

**Physicochemical characterization of nanoparticles.** Hydrodynamic radius and zeta potentials were determined by dynamic light scattering 1:10 diluted in PBS using a ZEN3600 Zetasizer Nano ZS (Malvern Panalytical, UK) at 37 °C. To quantify the number of ligands bound per AuNP, the PEG layer was released from the particles by dithiothreitol (0.1 M final concentration) at 37 °C for 15 min and at RT for 30 min analogously to the bio-barcode assay as described by Thaxton et al.[59]. AuNPs were removed from solution by centrifugation (45 min, 20817 rcf, 4 °C). The concentration of AuNPs after purification and azide-Fluor 488 in the supernatant after release was determined by UV–Vis spectroscopy (Lambda 25, Perkin-Elmer, USA).

**Production of titin-based tension probes.** Tension probes based on the I27 domain of titin were expressed in BL21 chemocompetent E. coli (#C2530H, NEB, USA) as previously described[20] from the plasmid pET22b-I27-RGD/E for the modified I27 domains containing an RGD or an RGE sequence for specific integrin adhesion or no specific adhesion, respectively, and the amber codon (TAG) as well as the pEVOL-pAzF plasmid for p-azidophenylalanine incorporation at the amber codon. Proteins were purified by immobilized metal ion affinity chromatography with a 1 ml His-Trap column (Äkta pure system, GE healthcare, UK) with buffer A: KH₂PO₄ buffer (pH 7.4) + 1 mM DTT + 30 mM imidazole and a gradient of 0–100 % buffer B: KH₂PO₄ buffer (pH 7.4) + 1 mM DTT + 500 mM imidazole. Purified proteins were desalted by gel filtration with Zeba spin columns (7000 MWCO, Thermo Fischer Scientific, USA) and subsequently labeled with ten-fold molar excess of Alexa647-NHS (Thermo Fischer Scientific, USA) in 0.1 mM KH₂PO₄ buffer at RT overnight. Unbound dye was removed again by gel filtration. Protein and dye concentration were quantified by UV–Vis absorption (NanoDrop ND-1000, Peqlab, Germany). Degree of labeling was determined to be 1.2, hence the majority of titin-based tension probes bears 1 fluorophore.

**Substrate modification for binding of tension probes.** Glass substrates were initially passivated with a covalently attached, self-assembled monolayer of Silane-PEG. This prevented non-specific adsorption of the tension sensors to the background. Thiol and alkyne-modified Silane-PEG derivatives allowed to immobilize AuNPs for the tension probes and cRGD-azide as ligand for cell adhesion, respectively. Briefly, glass coverslips (20 × 20 mm², #1, Carl Roth, Germany) were washed and activated in freshly prepared piranha solution (3:1 H₂SO₄/H₂O₂ (30 %)) for 1 h, rinsed three times in MilliQ water, sonicated for 5 min and dried with compressed nitrogen. (CH₃CH₂O)₃Si-PEG-methoxy (MW ~2000 g mol⁻¹, Biochempeg, USA), (CH₃CH₂O)₃Si-PEG-lipoic acid (synthesized from lipoic acid-PEG-NH₂ (MW ~3400 g mol⁻¹, Biochempeg, USA) and 3-(triethoxysilyl)-propyl isocyanate (Sigma-Aldrich) in 3 ml dry N,N-dimethylformamide (p.a., Carl Roth, Germany) similarly to (CH₃CH₂O)₃Si-PEG(2000) as previously described[60] and (CH₃CH₂O)₃Si-PEG-alkyne (MW ~3300 g mol⁻¹, Rapp Polymere, Germany) were solved in dry toluene (dried over 3 Å molecular sieves) in a 35:4:1 molar ratio with total silane concentration >125 μM to ensure complete coverage of the glass surfaces. Coverslips were immersed in this solution and the reaction was catalyzed by 25 μM triethylamine (≥99.5%) and left under nitrogen atmosphere at 80 °C for 16 h. The substrates were subsequently washed in ethyl acetate (p.a., AppliChem, Germany) and methanol (p.a., Carl Roth, Germany) for 5 min each by ultrasonication and dried with compressed nitrogen. To allow cell adhesion to the substrates, cRGD-azide was clicked to the substrates at 150 μM end concentration and incubated for 2 h. Subsequently, coverslips were washed in PBS, dried and 300 μl of gold nanospheres (9 ± 2 nm, 0.05 mg ml⁻¹ in tannic acid, nanoComposix, USA) were incubated at RT for 30 min Unbound nanospheres were washed off under a stream of MilliQ water. Samples were immediately incubated with 300 μl of the Alexa647-I27-RGE or Alexa647-I27-RGD tension probe (60 nM final concentration for cRGD ligands or 3 nM for reovirus) mixed with COOH-PEG₈-SH (19.2 nM final concentration) in 0.1 M KH₂PO₄ buffer (pH 7.4) at RT for 1 h. Note, that to enable observation of signals from individual tension sensors with virus ligands, we reduced their concentration by 20-fold compared to the experiments with cRGD ligands. All samples were glued onto 35 mm polystyrene Petri dishes with home-made 18 mm in diameter cut-outs by TwinSil (Picodent, Germany) and washed with 50 ml sterile PBS carefully avoiding to completely dry the surface at any time.

**Covalent immobilization of ligands on tension probes.** cRGD-alkyne (Cyclo (Arg-Gly-Asp-D-Phe-Pra) Biotrend Chemikalien GmbH) was clicked at 150 μM final concentration in the control experiments to the tension probes in absence of cRGD on the PEG background of the sample. Alkyne-modified reovirus particles were clicked in all other cases to the tension probes at a final concentration of

~$5 \times 10^9$ particles per ml. Reaction conditions were adopted from Hong et al.[58] as described above.

**Non-covalent immobilization of viruses or nanoparticles**. Biotinylated reoviruses or AuNPs were noncovalently immobilized via neutravidin (A2666, Thermo Fischer Scientific, USA), which is sandwiched on a biotin layer. Therefore, glass coverslips were passivated with 100% Silane-PEG-alkyne in toluene at 80 °C overnight as described above. After washing, 150 µM cRGD-azide, 150 µM azide-EG₃-biotin (Jena Bioscience, Germany), 500 µM THPTA, 100 µM CuSO₄, 5 mM aminoguanidine and 5 mM sodium-L-ascorbate were mixed in 100 mM phosphate buffer (pH 7) and glass substrates were inverted on a 100 µl drop of this reaction solution at RT for 2 h. Samples were washed subsequently under a stream of MilliQ water and carefully dried by draining. Neutravidin was reconstituted at 5 mg ml⁻¹ in PBS and stored in 10 µl aliquots at −20 °C. Solutions were diluted to a final concentration of 50 µg ml⁻¹ in PBS and a drop of 150 µl was placed on top of the biotinylated glass surfaces in a humid chamber at RT for 1 h. Afterwards, samples were washed three times with PBS for 5 min to remove unbound neutravidin. Successful binding of the protein was indicated by a visible change in hydrophilicity after careful drying by draining. 100 µl of the biotinylated reovirus or AuNPs with a final concentration of ~$1 \times 10^{10}$ particles per ml were directly added to the remaining liquid film and allowed to interact with the neutravidin in the humid chamber at RT for 2 h. Samples were glued onto 35 mm polystyrene Petri dishes with home-made 18 mm diameter cut-outs by TwinSil (Picodent, Germany) and washed with 50 ml sterile PBS avoiding to completely dry the surface at any time.

**Cell culture**. All cell culture media and supplements were purchased from Thermo Fischer Scientific unless otherwise stated. BSC1 cells stably expressing σ2-eGFP (small subunit of AP-2 fused to eGFP)[61], were used in experiments with molecular tension probes as they are a model system for reovirus infection[57]. HeLa cells (ATCC, CCL-2) were used for virus and nanoparticle uptake as frequent model for cellular uptake of nanoparticles[34,62] and receptor-specific uptake of reovirus[9,10]. Rat embryonic fibroblasts stably expressing paxilin-YFP (REF, originally obtained from Benny Geiger, Weizman Institute, Israel) were used as control for high magnitude integrin-mediated forces on the molecular tension probes with RGD ligands to compare to previous results[20]. Wildtype U373-MG (a kind gift from Tomas Kirchhausen, Harvard Medical School, Boston, USA) and U373-MG cells stably expressing the σ2-eGFP[21]. All adherent cell lines were cultured in DMEM (high glucose) supplemented with 10% FBS and 1% penicillin-streptomycin and incubated at 37 °C with 5% CO₂ and passaged with StemPro Accutase. L-cells for virus production were cultured in Joklik MEM medium (Sigma-Aldrich) supplemented with 1% L-Glutamine, 2% FBS, 2% Neonatal calf serum and 1% penicillin-streptomycin at 35 °C. To block β₁ integrins, 10⁵ HeLa cells were incubated in 200 µl serum-free media equipped with human integrin β₁ (CD29) blocking antibody clone P5D2 (#MAB117781, R&D systems, USA) at 10 µg ml⁻¹ final concentration for one hour before seeding. All experiments were conducted in 3 ml total DMEM medium containing 10% FBS and cells were allowed to settle for 1 h or 30 min before starting the time-lapse microscopy of titin-based tension probes and biotin-linkers, respectively. Actomyosin blocking experiments were performed by addition of drugs 1 h post seeding cells at a final concentration of 60 nM CytochalasinD (#C8273), 100 nM LatrunculinA (BML-T119, Enzo Life Sciences, USA) or 20 µM ROCK inhibitor Y-27632 (ALX-270–333, Enzo Life Sciences, USA), all previously diluted in 1 ml media.

**Indirect immunofluorescence**. Cells were fixed with 4% (w/v) paraformaldehyde in PBS for 20 min Samples were washed twice in PBS and for membrane permeabilization cells were treated with 0.5% (v/v) Triton X-100 for 10 min Virus factories were detected with an antibody against the non-structural reovirus µNS protein (anti µNS rabbit IgG, 1:200, generated by GenScript, USA) and secondary antibody Alexa Fluor 488 goat anti-rabbit IgG (1:200, Thermo Fischer Scientific, USA).

**Critical point drying and scanning electron microscopy (SEM)**. Samples were fixed in 2% glutaraldehyde in PBS for 15 min Water was stepwise replaced with ethanol (25%, 50%, 75%, two times 95% and three times 100%, each 10 min) and then critical point dried in a CPD 030 (Bal-Tec, Lichtenstein). Dried samples were sputter-coated with carbon (ACE 200, Leica, Germany) and imaged by SEM (Leo1530, Zeiss) with in-lense detector and 5 kV acceleration voltage at working distances between 9 and 11 mm.

**Wide-field fluorescence microscopy**. Wide-field fluorescence, interference reflection microscopy (IRM) and molecular tension fluorescence microscopy (MTFM) images were acquired on a DeltaVision system (GE healthcare, UK) based on an Olympus IX inverted microscope (Olympus, Japan) equipped with a cooled CCD camera (Coolsnap HQ, Roper Scientific, USA) and a home-build environmental chamber for temperature and CO₂ control. For imaging a 60x/1.4 NA PlanApo oil immersion objective lens (Olympus, Japan) was used.

**Confocal imaging**. Fixed and live cells interacting with single virus particles were imaged using the 405, 488, 561 and 633 nm laser lines of an inverted confocal microscope (LSM880, Zeiss, Germany). For live-cell imaging, a stage-top incubator (PM 2000 RBT, PeCon, Germany) and for imaging the LD C-Apochromat 40x/1.1 NA water immersion objective lens (Zeiss, Germany) was used. To limit the phototoxic effect of the laser light for living cells, images were recorded with low laser intensity of the 633 nm laser and brightfield images were obtained from the same scans with the transmitted light detector (T-PMT). To follow tearing of biotin-NeutrAvidin immobilized virus particles from glass surfaces, cells were seeded directly at the microscope and allowed to settle for 15 min. Up to 10 positions with adhering cells were chosen and imaged from 25 min post seeding on with images taken every 5–10 min for up to 16 h.

**TIRF imaging**. Single-molecule imaging of molecular tension probes at low density in combination with virus particles as ligands was performed on a commercial TIRF microscope (Nikon Eclipse Ti) equipped with a TIRF illumination unit, an autofocus system (PFS2) and a 100x 1.49 NA oil immersion objective (Apo TIRF, all Nikon, Japan)". The 488, 561 and 640 nm laser lines of a fiber-coupled multi laser engine (MLE-LFA, TOPTICA Photonics, Germany) were used for illumination. Excitation light was filtered using a quadband notch filter. Excitation and emission light were separated using a quadband dichroic mirror (all filters AHF Analysetechnik, Germany). Emitted light was filtered further using 525/50 nm or 605/70 nm bandpass or 640 nm longpass filters mounted in a three-way image splitter (Optosplit II, Cairn Research, Faversham, United Kingdom). Data was acquired with a back-illuminated emCCD camera (iXon Ultra 897, Andor, Northern Ireland). Live-cell imaging was performed at 37 °C using a custom-built stage-top heating chamber and an objective heating collar. Images for individual channels were recorded consecutively with 100 ms exposure time at 0.3 Hz frame rate per channel triplet. Illumination intensities of 1.3 mW at 488 nm, 1.7 mW at 561 nm and 2.4 mW at 640 nm as measured at the position of the back focal plane of the objective were used for all experiments. Time-lapse Supplementary Movies were recorded 1 h post seeding of the cells for 10 min per. TetraSpeck™ multi-color fluorescent microspheres (0.1 µm, blue/green/orange/dark red, Thermo Fischer, USA) were used used as reference to align spectral channels by projective transformation using custom-written Matlab® code (MathWorks, USA).

**Image processing and data analysis**. Fluorescent images were processed using Fiji software (Fiji is just ImageJ, version 1.51n, http://imgej.nih.gov/ij). Brightness, contrast and LUTs of microscopy images were adjusted for the presentation.

**Molecular tension fluorescence microscopy analysis**. For the analysis of fluorescent signals from single molecular tension probes, TIRF images were analyzed with a custom-written program[63] in Matlab (MathWorks, USA), which was kindly provided by Kristin S. Grußmayer (EPFL, Switzerland). First, a binary mask from the maximum projection of the virus particles was created by a particle tracking algorithm. Traces of the fluorescence intensity over time were calculated from the sum of a 3-pixel wide radius at each particle. The local background was calculated from a 3-pixel wide annulus starting at 5 pixels from the spot center and subtracted from the sum intensity. Background-corrected fluorescence traces of the tension probe channel were manually evaluated for events showing an increase in fluorescence, which is corresponding to the opening of a tension sensor. Additionally, a mask of the cells was manually created from the fluorescence signal in the AP2-GFP channel. Events were subsequently categorized to be either included in the projected cell area or in the background outside of cells. For events occurring in absence of virus particles, random analysis spots were created by rotation of the virus channel at multiple of 90° angles. These spots without viruses were further processed as described above and categorized under the projected cell area or in the background.

**Analysis of removal of immobilized viruses and nanoparticles**. Confocal images of fluorescently labeled viruses and nanoparticles were corrected for photo bleaching and drift using the Fiji plugins bleach correction with background = 0 and MultiStackReg with rigid body transformation. Pixel classification and automated tracking was conducted by a machine learning algorithm in ilastik 1.3.0 (http://ilastik.org). Projected cell areas were manually traced from fluorescence transmission images in Fiji. Number of stable particles in the ROI were analyzed from the tracking results using a processing script in Python as follows:

In the simple analysis with one ROI, all particles underneath the projected cell area 1 h post seeding were considered for the analysis (Fig. 3b). This underestimates the relative number of torn off viruses in the first frames, when the cell did not yet cover the whole area or in later frames, when cells are migrating outside the area. However, due to variability between the cells and the different conditions tested, this proved to be the most robust approach. The remaining particles in this area were normalized over time to those particles present in the first two frames. Thereby any tearing events, that happened during the first 30 min after cell seeding were neglected. We define $t_0$ as 30 min post seeding regardless of the spreading phase of individual cells.

Alternatively, we analyzed the number of particles in multiple ROIs (Fig. 3b). This allows to account for different interaction times of cells with particles during spreading.

However, it requires tracing of the projected cell area in each frame. Particles that disappeared during the time-lapse (ripped off ($t^*_n$)) were traced back for how long they have been within the projected cell area (interaction time $t^*_n$) and normalized against all particles that theoretically could have been ripped off at that time point (total ($vt^*_n$)) resulting in %ripped off ($t^*_n$) = (ripped off ($t^*_n$)/ total ($t^*_n$)). For the initial spreading area at 25 min post seeding we chose $t^*_2 = 5$ min since spreading was observed in most cells after 20 min Note, that one has to take into account those particles for total ($t^*_n$) that had been already torn off at all shorter interaction times, since the exponential decay requires a normalization to the initial value. We solved this issue by dividing the absolute number of particles that stayed within the projected cell area for at least $t^*_n$ (available ($t^*_n$)) by $1 - \sum_{i=0}^{n-1} (\% \text{ ripped off}(t^*_i))$ for $n > 0$ and defined total ($t^*_0$) = available ($t^*_0$) for $n = 0$ in an iterative calculation. Finally, we plotted the relative number of particles that stayed underneath the cell area for each $t^*_n$ as % stayed ($t^*_n$) $= 1 - \sum_{i=0}^{n} \% \text{ ripped off}(t_i)$.

To validate the two methods, we analyzed three test data sets with both approaches (Fig. 3c, d). Since the results did not significantly chance we used the simple analysis with one ROI for all further results.

Relative numbers of remaining particles under $n$ cells were plotted as mean ± s.e.m. for analysis of the kinetics in Prism 7 (Graphpad Software). We fitted a two-phase decay function until $t = 70$ min (Eq. 5) with normalized initial concentration of virus particles $c_{\text{particles}}(t_0) = 1$, final concentration $c_{\text{particles}}(t_{\text{end}}) = 0$ (since complete uptake was observed after overnight incubation) and $k_{\text{off}}^0$ as determined in a one-phase exponential decay in a single ROI, for viruses/particles in the background without cells. For presentation of variability between cells, the fitting parameters $k_{\text{off}}$ and $a$ were plotted from individual fits per cell in Figs. 3, S6 and S8 excluding these fits that were either hit-constraint or ambiguous.

**Statistics**. All experiments were conducted as three technical replicates. All tests were performed in Prism 7 (Graphpad Software) and data presented as mean ± s.e.m. or boxplots as median ± 95% confidence interval with whiskers representing the min and max values.

**Reporting summary**. Further information on research design is available in the Nature Research Reporting Summary linked to this article.

## Data availability

The data that support the findings of this study are available from the corresponding authors upon request. A reporting summary for this Article is available as a Supplementary Information file. The source data underlying Figs. 1d, 2d, e, 3c, d, 4c–e, 5b–d and Supplementary Figs. 2b, 3b, d, 5b, d, e, 6 and 7a–c are provided as a Source Data file.

## Code availability

Matlab script for extraction and analysis of single particle traces from TIRF images and Python script for analysis of spatially stable particles in ROIs from tracking results are publicly available on https://doi.org/10.5281/zenodo.3556565, https://doi.org/10.5281/zenodo.3556563 and https://doi.org/10.5281/zenodo.3551377.

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

## Acknowledgements

This work was supported by the German Science Foundation (SFB1129 project P15 to E.A.C.-A. and J.P.S., P14 to S.B. and P4 to U.S.S., EXC81 to D.-P.H.), the BMBF/VDI (MorphiQuant3D to D.-P.H.) and the Max Planck Society. T.W. was supported by the Boehringer-Ingelheim Foundation. F.F. was supported by the Heidelberg Graduate School for Fundamental Physics (HGSFP). We thank Kristin S. Grußmayer (EPFL, Switzerland) for sharing the Matlab code for image analysis of single particles, Kota Miura (Network of European Bioimage Analysts) for his work on the image analysis software on single particle tracking and Florian Rechenmacher and Horst Kessler (TU Munich, Germany) for providing the integrin-selective ligands. We thank the members of the J.P.S., E.A.C.-A., and S.B. laboratories for technical assistance and discussions.

## Author contributions

E.A.C.-A. and J.P.S. conceived the study, T.W., Y.L., K.S., S.B., J.P.S., and E.A.C.-A. designed the experiments, T.W., K.Y., K.G., Y.L., M.F., J.O., and E.W. performed the experiments, F.B. provided software, F.F. and U.S.S. carried out the theoretical modeling, and T.W. wrote the manuscript. All authors commented on the manuscript and contributed to it.

## Competing interests

The authors declare no competing interests.
