## [Peer Review File · Nature Communications]

Reviewers' Comments:

Reviewer #1:

Remarks to the Author:

In this study by Wiegand et al., force magnitudes required for the uptake of reovirus particles by cells are analyzed. The authors adapted a previously published molecular tension probe that is based on the I27 domain of titin and was reported to monitor comparably high mechanical forces (based on independent AFM measurements, the probe is expected to sense forces of about 80-200 pN). The authors coupled virus particles to surface-immobilized I27 tension sensor probes and studied virus uptake forces in cell culture.

In a second approach, the authors functionalized reovirus particles with neutravidin and non-covalently immobilized them on biotin-activated surfaces; they estimated multiple neutravidin-biotin bonds per virus particle in this configuration. When the authors used these surfaces in cell culture experiments, they found that cells are able to tear off virus particles from the surface. By calculating virus particle off-rates and applying a simple Bell's model they estimated mean forces of about 21-22 pN per virus particle.

Next, the authors mimicked virus particle uptake by immobilizing functionalized gold nanoparticles. As expected, also these particles were torn off by cells and the results indicated that this process is enhanced by the presence of integrin-ligands on the gold particle. The data also show that particle uptake can occur in the absence of specific ligand-cell interactions indicating a complex mechanisms for virus uptake. Finally, the energetic cost for particle uptake was estimated.

Overall, this is an interesting report. The authors established a range of technically challenging experiments and they used the tools to investigate a biologically interesting question. However, the quantitative conclusions are not convincingly supported by the data, and the biological insights gained from this study are limited.

Major points:

1. Based on the I27 experiments, the authors conclude that virus uptake forces are exceeding 40 pN and they argue that this value is in the range of interactions strengths that were previously reported for the binding of other viruses to cells. The conclusion is somewhat misleading, because the I27 probe was reported to sense significantly higher force ranges. Previous studies showed that the I27 domain unfolds at forces of 80-200 pN. In fact, the original study (Galior et al., 2016) specifically emphasized that the I27-based probe can monitor exceptional high forces that are not accessible with other techniques like the DNA gauge tether from the Ha lab (which was used to measure forces of about 40 pN).
2. In the second part of the study, the authors immobilize virus particles on a surface through a non-covalent neutravidin-biotin linkage. They then use Bell's Model to calculate the mean force per particle from the apparent off-rates and estimate forces of about 21-22 pN per particle. They conclude that these values are in agreement with the results obtained by the I27 probe. It is questionable, however, that the Bell's Model approach can be used to reliable estimate forces in this context, because the assumption that cells pull on the virus particle with a constant force is most certainly invalid. The authors estimate in later experiments that about 35 cell-virus interactions can be expected per particle, which should lead to a rather complex ensemble of forces. It should be also noted that the unbinding force, even for a simple neutravidin-biotin linkage, is expected to be loading rate dependent, but the loading rates are also unknown. As a result, the precision of these particular force calculations is uncertain and it remains unclear how high the force per virus particle really are.
3. Irrespective of the uncertainties about the actual force magnitudes underlying the virus-cell interaction, novel biological insights are missing. The role of integrin receptors for reovirus uptake has been documented before, and it is to be expected that blocking integrin activity impairs the ability of cells to tear off virus particles from the surface. Thus, the biological relevance of the study is rather low.

Additional points:

The authors state that treatment of cells with a MMP inhibitor does not affect virus particle detachment. This is a crucial control and requires quantification (in addition to showing a movie and a few pictures). The authors also report that treatment of cells with $\beta 1$ integrin blocking antibodies impairs virus uptake, but they also show that it leads to a reduction in projected cell area. Could the decrease in virus particle uptake not be a direct consequence of reduced cell area?

Reviewer #2:

Remarks to the Author:

In this article the investigators look at cell uptake of viral and gold particles with specific ligands from the ventral surface of cells. There is a large non-specific component in that passivated gold particles are taken up by the cells at a high rate, which means that the specific binding energy is only about 20% of the total binding energy of the particles. Further, there is little new here because the values of the forces needed to break bonds are derived from ensemble average measurements and hence approximate. The basis of force generation is not tested. Because the forces are similar to the values obtained from the dorsal surface, they are believable. Thus, although this is a nice melding of different technologies to look at the forces involved in endocytosis, there are several major weaknesses in the current manuscript.

1. Does the force for endocytosis come from myosin or from membrane-associated proteins? They can simply add a myosin or actin inhibitor to see if that affects the rate of endocytosis. To alter membrane properties, they can alter membrane trafficking with brefeldin A.
2. Is non-specific uptake of gold particles altered by passivating the particles differently? With a truly passivated gold particle, the specific binding energy of ligands to receptors would be a much greater fraction of the total binding energy.

Particularly, point 1 needs to be addressed to increase the novelty of the paper.

Reviewer #3:

Remarks to the Author:

In this study, a single molecule tension sensor based on titin I27 domain and NSET fluorescence quenching was developed to measure force exerted by cells during the uptake of Reovirus particles from the ventral cell surface. Live-cell TIRF imaging reveals that the sensor, when covalently coupled, report force of >40 pN, consistent with the force range reported by other methods. It was also shown that non-covalent conjugation such as by biotin-neutravidin can be ruptured by cellular force, which allows cell to internalize the virus and become infected. By quantitatively analysis, the mean force of ~ 20 pN per particle was obtained. The author then functionalized nanoparticles with specific adhesion motif. Here they observed that integrin $\alpha 5\beta 1$ specific ligands have a strong effect in promoting particle uptake. However, the roles of other adhesion receptors seem to also be implied. Mathematical model was developed to calculate the adhesion energy, quantifying that the virus adhesion energy is higher than for the nanoparticles.

Overall, this is an interesting and solid technical development. The experiments are well-described and supported by theoretical analysis. There are a number of suggestions in terms of presentation and organization below.

1. While the experiments are technically well-designed and well executed, how this technique may inform physiological processes may need to be motivated further. In particular, the scenario in which viral infection occurs through the ventral side would seem to be quite limited as tissues are normally lined with polarized epithelia that have robust barrier function. One would think that the virus uptake through the luminal (dorsal/apical) side of polarized epithelia would be the predominant mode of viral entry. Furthermore, for epithelia, the ventral side of the cell will be closely apposed to the basal lamina and it is difficult to imagine how virus particles would insert

itself into that space.

This, in my opinion, is one of the conceptual limitation of this study. But if the authors could discuss these issues in more details, and motivated by examples of ventral-side viral infection, then this concern would be mitigated.

2. Related to #1, if the authors could highlight new biological insights uncovered here, that will strengthen this manuscript substantially. On the one hand, the authors mentioned that the force range obtained is similar to the dorsal side measurements in previous studies, which could be expected. On the other hand, the authors noted that the cells have the spatial preference in the uptake the particles at cell edge or under the lamellipodia, that could be interesting to expand upon. Also, is this a clathrin-dependent or independent process?

3. The possibility of multi-valent interaction is mentioned. If it is possible to combine stoichiometry of receptors with force measurement, this could strengthen the technical aspects of this study even further.

4. In the current form, the article is very condensed with only 3 main figures. While well-written, it is very compact and takes some effort to follow. It would be helpful to expand the results to more figures. For example, more results as well as the data analysis and mathematical modelling could be illustrated in additional main figures.

Reviewer #4:

Remarks to the Author:

The study by Tina Wiegand et al. analyzes the uptake of viruses and nanoparticles from the ventral side of cells. Indeed, this aspect of viral entry is novel and expands the repertoire of methods to study mechanical forces during uptake of particles.

The techniques established in this study contribute very significantly to the rapidly growing toolbox, which is needed to characterize virus and nanoparticle uptake from the ventral side of cells. However, further controls and statistical analysis are recommended to validate the titin I27-based tension sensors.

Insights from coupled virus particles and nanoparticles allowed the authors to gain new information on the interaction rates and to estimate the number of receptor-ligand bonds necessary to provide the calculated additional adhesion energy. The image analysis could optionally be further optimized, by including particle tracking, and open questions still exist concerning the off-rate of virus particles in comparison to nanoparticles.

In summary: This is a very good manuscript, dealing with a timely topic, well written and thoroughly discussed.

Recommendation: Publish after revision.

Specific comments and questions:

PartI: Measurement of force acting on tethered virus particles using NSET titin I27-based tension sensors

In the first part of the study, a nanometal surface energy transfer (NSET) titin I27-based force sensor was used to estimate the force acting on single virus particles. The authors employed a sophisticated coupling strategy to attach gold nanoparticles via the force sensor to the virus particle. All necessary controls for the production and characterization of tension sensor probes as well as immobilization of nanoparticles have been performed. A fluorescence signal caused by unfolding of titin I27 domains could be observed and was correlated it with the position of virus particles. However, even for tension sensors not engaged with a virus or in the background

significant titin, unfolding was measured. Fig. 1d shows the data for 10 cells with engaged virus particles. A closer reading of the Supplementary material showed, that only 114 events from 1073 virus particles are part of these data points. Therefore, the statistical values, which are given in the figure legend are misleading and should be better explained. For the background signal only 36 and 39 data points were measured from 9 and 11 cells, respectively. With such a low number of events per cell it is not representative to calculate mean values for each cell. The authors should also explain why the supplementary Movie 1 shows a decrease in signal over time.

It remains unclear whether the force per virus particle can be deduced from this data. Since this point is of central importance for the paper the authors should explain in greater detail how the number of sensors interacting with each virus particle can be resolved with their approach.

The authors argue, that multiple sensors increase the mechanical resistance and thus 40 pN is a lower estimate for the force. However, the fluorescence signal from which the average unfolding rate is calculated, might also be caused by two or more uncorrelated sensor modules. A higher number of engaged sensor modules per particles could for example be caused by the spatial confinement of viruses below the cell, which might favor binding to multiple sensors.

In addition, tension is inherently overestimated, as tension sensors, which do not show a signal in the observation period of 10 min, are not included in the calculation. With an average rate of $k_{\text{off}} = 1.3 \times 10^{-4} \text{ s}^{-1}$ only 1.6 events are expected in each 10 minutes interval!

Ultimately the additional use of either very stiff or very soft sensor modules as well as tension sensors with a high or low zero-force unfolding rate would be recommended. However, this suggestion goes beyond the scope of this manuscript.

For a better understanding of the data analysis procedure the number of unfolding/refolding events per time trace would be recommended.

Part II: Uptake of biotin-neutravidin tethered virus particles and nanoparticles

In the next step, the authors proceed to a set-up where virus particles and nanoparticles are no longer covalently linked to the surface but can be taken up by the cell upon rupture of the biotin-neutravidin link. Comparing virus particles with nanoparticles coated with different ligands is a good approach to dissect biophysical mechanisms behind virus uptake. The term "virus-like" nanoparticles, which is used in the title is confusing and it would be better to call them "virus-sized".

The amount of virus particles directly below the cell area decreases over time compared to background densities. This is clearly demonstrated by the data and the uptake is faster in lamellipodial regions. From the relative k_{off} rates of background and cell area, an average force on the virus particles is calculated using the Bell model. This should indeed give an approximation of the forces involved in virus particle uptake.

Next, the nanoparticles with and without specific ligands are characterized. It remains unclear why the background off-rate, which is used for the double-exponential fit of data from nanoparticles was not taken into account for the fit of virus particles. It also remains to be discussed why the apparent off-rate in absence of cells is different for virus particles and nanoparticles ($k_{\text{off}} \text{ Virus} = 7.2 \times 10^{-6} \text{ s}^{-1}$ vs. $k_{\text{off}} \text{ AuNP} = 12.2 \times 10^{-6} \text{ s}^{-1}$) and how this influences comparison of relative forces between them. From the movies it seems that a subset of the particles is also pushed away by the cell rather than being taken up. Especially for nanoparticles without a specific ligand AuNP(1) many particles are dragged by the cell. Automated tracking of particles instead of only determining their position would be necessary to optimize the analysis and either exclude these events or analyze them in more detail. Another way to optimize the set-up – which is beyond the

scope of the current study - could be the use of monovalent forms of avidin and a lower surface density to ensure single particle interactions.

We now resubmit our manuscript with the modified title “Forces during cellular uptake of viruses and nanoparticles at the ventral side” for publication in Nature Communications.

We appreciate the positive feedback and constructive criticism the reviewers have given. In fact, their comments have strongly helped us to improve the manuscript. With the incorporation of new experimental data on the impact of actin/myosin on nanoparticle uptake (Fig. 5) and a stronger focus on the spatial preference of uptake (Fig. 3) as well as the resulting implications for the hypothesized model (Fig. 6), we have thoroughly edited the text and expanded on the biological insights as suggested by the reviewers.

We want to point out that following the suggestion of reviewer #4, we now immediately present the rates and force estimates of tearing off biotin-neutravidin immobilized viruses from two-phase exponential fits analogously to the analysis of the nanoparticles and summarized these force values in the abstract to ~30 pN.

Our main findings remain unchanged: forces during particle uptake from the ventral cell side are in the same order as reported for dorsal interactions. However, they can be tuned by close cell-matrix contacts and specific adhesion ligands on the particles providing enough energy for the detachment of surface-immobilized particles. Actomyosin inhibition strongly impairs cellular attachment and thus significantly lowered the chance of particle uptake, but we do not find a significant change in uptake kinetics.

Reviewer #1 (Remarks to the Author):

In this study by Wiegand et al., force magnitudes required for the uptake of reovirus particles by cells are analyzed. The authors adapted a previously published molecular tension probe that is based on the I27 domain of titin and was reported to monitor comparably high mechanical forces (based on independent AFM measurements, the probe is expected to sense forces of about 80-200 pN). The authors coupled virus particles to surface-immobilized I27 tension sensor probes and studied virus uptake forces in cell culture.

In a second approach, the authors functionalized reovirus particles with neutravidin and non-covalently immobilized them on biotin-activated surfaces; they estimated multiple neutravidin-biotin bonds per virus particle in this configuration. When the authors used these surfaces in cell culture experiments, they found that cells are able to tear off virus particles from the surface. By calculating virus particle off-rates and applying a simple Bell's model they estimated mean forces of about 21-22 pN per virus particle.

Next, the authors mimicked virus particle uptake by immobilizing functionalized gold nanoparticles. As expected, also these particles were torn off by cells and the results indicated that this process is enhanced by the presence of integrin-ligands on the gold particle. The data also show that particle uptake can occur in the absence of specific ligand-cell interactions indicating a complex mechanisms for virus uptake. Finally, the energetic cost for particle uptake was estimated.

Overall, this is an interesting report. The authors established a range of technically challenging experiments and they used the tools to investigate a biologically interesting question. However, the quantitative conclusions are not convincingly supported by the data, and the biological insights gained from this study are limited.

Major points:

1. Based on the I27 experiments, the authors conclude that virus uptake forces are exceeding 40 pN and they argue that this value is in the range of interactions strengths that were

previously reported for the binding of other viruses to cells. The conclusion is somewhat misleading, because the I27 probe was reported to sense significantly higher force ranges. Previous studies showed that the I27 domain unfolds at forces of 80-200 pN. In fact, the original study (Galior et al., 2016) specifically emphasized that the I27-based probe can monitor exceptional high forces that are not accessible with other techniques like the DNA gauge tether from the Ha lab (which was used to measure forces of about 40 pN).

Indeed, the forces might be higher with ~40 pN being only a lower estimate, while the I27 probe can withstand higher forces. However, in the original report (Galior et al., 2016) as well as in previous studies using AFM to induce mechanical unfolding of I27 (Li et al., 2000, DOI: 10.1038/81964) the loading rate dependence was noted and thus also smaller forces can be sufficient when applied over a longer period of time, i.e. lower loading rates. Here, to estimate the minimal force applied by the cells, we assumed that a constant force was acting on the viruses over the whole period of the experiment and used the Bell model to compare the off/unfolding rates induced by the cells vs in a control region and found a consistently small value of 44 pN. We emphasized the loading rate dependence in the revised manuscript and moved the supplementary calculation based on Bell's model into the main text line 129-151.

2. In the second part of the study, the authors immobilize virus particles on a surface through a non-covalent neutravidin-biotin linkage. They then use Bell's Model to calculate the mean force per particle from the apparent off-rates and estimate forces of about 21-22 pN per particle. They conclude that these values are in agreement with the results obtained by the I27 probe. It is questionable, however, that the Bell's Model approach can be used to reliably estimate forces in this context, because the assumption that cells pull on the virus particle with a constant force is most certainly invalid.

Since we have no data on the actual force history and the loading rates, our minimal assumption is a constant force. We certainly acknowledge the possibility of other force histories, which can furthermore vary between individual particles and stress this in line 137-141 in the revised manuscript. However, any of these would infer higher forces than our statistical analysis over all particles underneath the cells.

The authors estimate in later experiments that about 30 cell-virus interactions can be expected per particle, which should lead to a rather complex ensemble of forces.

We agree with reviewer 1 that the ensemble of many cell-virus contacts could give rise to a more complicated force distribution. However, our mathematical model describes the average force calculated by Bell's model. From this, in the end we estimate the number of bonds between particle and cell and our calculation is based on the assumption that every particle-cell contact contributes a similar energy to the uptake process. However, as the number of bonds is a result of our estimation we cannot use the number for the model.

We are well aware that the situation for single particles might be more complex and the number of virus-cell contacts could change. Thus, in the revised manuscript we now explicitly state that our model describes the average and neglects that different virus-cell contacts contribute different energies (line 437-438).

It should be also noted that the unbinding force, even for a simple neutravidin-biotin linkage, is expected to be loading rate dependent, but the loading rates are also unknown. As a result, the precision of these particular force calculations is uncertain and it remains unclear how high the force per virus particle really are.

We emphasized the loading rate dependence a couple of times in the manuscript. We agree that the absolute values strongly depend on the assumptions being made and the ensemble measurement only provides the average force per virus. Nonetheless, it allowed us to assess the lower limit of forces.

3. Irrespective of the uncertainties about the actual force magnitudes underlying the virus-cell interaction, novel biological insights are missing. The role of integrin receptors for reovirus uptake has been documented before, and it is to be expected that blocking integrin activity impairs the ability of cells to tear off virus particles from the surface. Thus, the biological relevance of the study is rather low.

The role of integrins for reovirus infection is not completely understood. While primary attachment is not impaired, reovirus entry was found to be reduced in beta1 knockout cells (Maginnis et al., 2006). They further reported a role of the NPXY motive in the cytoplasmic tail of beta1 integrins for the recruitment of the clathrin machinery (Maginnis et al., 2008). We, on the other hand show (by the usage of artificial nanoparticles mimicking the virus capsid ligands) that integrin blocking lowers the force but not the frequency of particle uptake. This aspect of mechanical contribution via e.g. adhesion energy was not previously discussed for reovirus, which we emphasized in the revised version line 464-466. Of note our observations do not interfere with the integrin-mediated recruitment of clathrin, however, integrin blocking did not alter forces on RGD coated nanoparticles but only on particles, which are integrin alpha5beta1-specific (see Figure S7).

Additional points:

The authors state that treatment of cells with a MMP inhibitor does not affect virus particle detachment. This is a crucial control and requires quantification (in addition to showing a movie and a few pictures).

We quantified the tearing of viruses before and after addition of the MMP inhibitor now for 14 cells in 2 independent data sets (Supplementary Fig. 5b) and confirmed that there was no significant difference compared to untreated control cells.

The authors also report that treatment of cells with b1 integrin blocking antibodies impairs virus uptake, but they also show that it leads to a reduction in projected cell area. Could the decrease in virus particle uptake not be a direct consequence of reduced cell area?

To exclude this possibility, we (i) corrected all data for the respective projected cell area and (ii) we note in the revised manuscript, that even though spreading was similarly reduced uptake of nanoparticles with RGD ligands was not affected while uptake of nanoparticles with alpha5beta1 ligands was (line 326-329).

Reviewer #2 (Remarks to the Author):

In this article the investigators look at cell uptake of viral and gold particles with specific ligands from the ventral surface of cells. There is a large non-specific component in that passivated gold particles are taken up by the cells at a high rate, which means that the specific binding energy is only about 20% of the total binding energy of the particles.

Further, there is little new here because the values of the forces needed to break bonds are derived from ensemble average measurements and hence approximate.

The basis of force generation is not tested. Because the forces are similar to the values obtained from the dorsal surface, they are believable. Thus, although this is a nice melding of different technologies to look at the forces involved in endocytosis, there are several major weaknesses in the current manuscript.

We thank the reviewer for the feedback. We agree that ensemble measurements give the mean value of all particles observed. While we have no data on the distribution of the individual forces the mean still gives valuable information and allowed us to calculate the contribution of the specific adhesion and to estimate the number of receptor-ligand interactions for virus particle uptake.

1. Does the force for endocytosis come from myosin or from membrane-associated proteins? They can simply add a myosin or actin inhibitor to see if that affects the rate of endocytosis. To alter membrane properties, they can alter membrane trafficking with brefeldin A.

We performed a range of different alterations of the biological system including affecting the membrane composition and endocytic pathways by Genestein and Cyclodextrin, Dynamin mutants, receptor blocking and actin/myosin inhibitors, which did not reveal any significance for the removal of surface-bound reoviruses in our preliminary data. Thus, we focused on the non-specific component of particle adhesion, which was observed with nanoparticles lacking specific ligands.

We now performed specific inhibitions of actin polymerization and ROCK during uptake of such nanoparticles and summarized the results in an additional main figure (5). We observed a general reduction in tearing of surface-bound particles, however, the rates remained similar suggesting that the forces do not come from actin/myosin (line 353-375). This could be explained by the high contribution of the specific and non-specific adhesion energy, which we discuss thoroughly in the revised manuscript (line 426-430).

2. Is non-specific uptake of gold particles altered by passivating the particles differently? With a truly passivated gold particle, the specific binding energy of ligands to receptors would be a much greater fraction of the total binding energy.

We agree with the reviewer that the passivation of the nanoparticles is reduced in our experimental set-up because of PEG modifications with ligands for surface-immobilization (biotin) and the organic dye for imaging. Hence, more biomolecules from the serum could adsorb and enhance the adhesion energy of the particles. However, this will be also true for the similarly immobilized and labeled viruses and might also play a role in vivo, where truly passivated particles do not exist as discussed in line 426-430. We thus took this number to compare the specific binding components in our system. Furthermore, we argue that the spatial confinement of particles in close vicinity to the cells facilitates non-specific interactions, which are else not sufficient to initiate and stabilize the contact with host cells in a soluble assay (line 334-337).

Particularly, point 1 needs to be addressed to increase the novelty of the paper.

Reviewer #3 (Remarks to the Author):

In this study, a single molecule tension sensor based on titin I27 domain and NSET fluorescence quenching was developed to measure force exerted by cells during the uptake of Reovirus particles from the ventral cell surface. Live-cell TIRF imaging reveals that the sensor, when covalently coupled, report force of >40 pN, consistent with the force range reported by other methods. It was also shown that non-covalent conjugation such as by biotin-neutravidin can be ruptured by cellular force, which allows cell to internalize the virus and become infected. By quantitative analysis, the mean force of ~ 20 pN per particle was obtained. The author then functionalized nanoparticles with specific adhesion motif. Here they observed that integrin $\alpha 5\beta 1$ specific ligands have a strong effect in promoting particle uptake. However, the roles of other adhesion receptors seem to also be implied. Mathematical model was developed to calculate the adhesion energy, quantifying that the virus adhesion energy is higher than for the nanoparticles.

Overall, this is an interesting and solid technical development. The experiments are well-described and supported by theoretical analysis. There are a number of suggestions in terms of presentation and organization below.

1. While the experiments are technically well-designed and well executed, how this technique may inform physiological processes may need to be motivated further. In particular, the scenario in which viral infection occurs through the ventral side would seem to be quite limited as tissues are normally lined with polarized epithelia that have robust barrier function. One would think that the virus uptake through the luminal (dorsal/apical) side of polarized epithelia would be the predominant mode of viral entry. Furthermore, for epithelia, the ventral side of the cell will be closely apposed to the basal lamina and it is difficult to imagine how virus particles would insert itself into that space. This, in my opinion, is one of the conceptual limitation of this study. But if the authors could discuss these issues in more details, and motivated by examples of ventral-side viral infection, then this concern would be mitigated.

While the main route of viral infection is certainly via the apical side, infections and physical damage are well known to disrupt the epithelial integrity opening access to all kind of pathogens. Especially for human papilloma virus the basement membrane is the primary site of virus binding and also for reovirus, as employed in this study, a basal preference has been shown in respiratory epithelial cells. We expanded on this point and explicitly named these examples in the revised manuscript in line 40-46.

2. Related to #1, if the authors could highlight new biological insights uncovered here, that will strengthen this manuscript substantially. On the one hand, the authors mentioned that the force range obtained is similar to the dorsal side measurements in previous studies, which could be expected. On the other hand, the authors noted that the cells have the spatial preference in the uptake the particles at cell edge or under the lamellipodia, that could be interesting to expand upon. Also, is this a clathrin-dependent or independent process?

We thank the reviewer for the suggestion and further highlighted that particle uptake can be triggered by the deformation of the membrane due to spatial confinement of particles at the ventral cell side in the revised manuscript. The spatial preference below lamellipodia can be linked to the particularly close distance to the substrate, which we illustrate by interference reflection microscopy in an additional main figure (Fig. 3a) and Movie 4. Quantitative

evidence for the spatial preference is given by the analysis with respect to the actual cell spreading area (with multiple ROIs) as explained in detail in line 205-213.

In earlier experiments with molecular tension probes we simultaneously imaged GFP-tagged clathrin adapter protein AP2 but did not observe a preference for colocalization of AP2 with respect to the viruses that show force signals. While reovirus is known to use multiple endocytic pathways (Schultz *et al.*, 2012, Journal of Virology), only about 15 % of likewise surface-immobilized virus particles interacted with the clathrin machinery in our previous study (Fratini 2018 ACS Infectious Diseases). We thus mention the contribution of clathrin in the manuscript but do not consider the process being clathrin-dependent. Furthermore, we experimentally tested the contribution of actin/myosin (Fig. 5) and hypothesize from the observations, that membrane deformation arising from close cell-matrix contacts and adhesion energy are sufficient for force generation as theoretically predicted by our mathematical model.

3. The possibility of multi-valent interaction is mentioned. If it is possible to combine stoichiometry of receptors with force measurement, this could strengthen the technical aspects of this study even further.

We thank the reviewer for this suggestion. As we do not have a direct read-out of the stoichiometry of cellular receptors actually interacting with the particles we modeled the adhesion energy as a function of the forces observed and calculated an average number of receptors thereof as now illustrated in the additional main figure 6.

4. In the current form, the article is very condensed with only 3 main figures. While well-written, it is very compact and takes some effort to follow. It would be helpful to expand the results to more figures. For example, more results as well as the data analysis and mathematical modelling could be illustrated in additional main figures.

We expanded the results to 6 main figures by implementing new results on the contribution of actin/myosin (Fig. 5) and expanding on the spatial preference (Fig. 3) and the mathematical modelling (Fig. 6). To improve clarity, we relocated supplementary data into individual figures (Fig. 3 & 6) and calculations to the main text (line 133-147, 218-227, 394-416).

Reviewer #4 (Remarks to the Author):

The study by Tina Wiegand et al. analyzes the uptake of viruses and nanoparticles from the ventral side of cells. Indeed, this aspect of viral entry is novel and expands the repertoire of methods to study mechanical forces during uptake of particles.

The techniques established in this study contribute very significantly to the rapidly growing toolbox, which is needed to characterize virus and nanoparticle uptake from the ventral side of cells. However, further controls and statistical analysis are recommended to validate the titin I27-based tension sensors.

Insights from coupled virus particles and nanoparticles allowed the authors to gain new information on the interaction rates and to estimate the number of receptor-ligand bonds necessary to provide the calculated additional adhesion energy. The image analysis could optionally be further optimized, by including particle tracking, and open questions still exist concerning the off-rate of virus particles in comparison to nanoparticles.

In summary: This is a very good manuscript, dealing with a timely topic, well written and thoroughly discussed.

Recommendation: Publish after revision.

Specific comments and questions:

PartI: Measurement of force acting on tethered virus particles using NSET titin I27-based tension sensors

In the first part of the study, a nanometal surface energy transfer (NSET) titin I27-based force sensor was used to estimate the force acting on single virus particles. The authors employed a sophisticated coupling strategy to attach gold nanoparticles via the force sensor to the virus particle. All necessary controls for the production and characterization of tension sensor probes as well as immobilization of nanoparticles have been performed. A fluorescence signal caused by unfolding of titin I27 domains could be observed and was correlated it with the position of virus particles. However, even for tension sensors not engaged with a virus or in the background significant titin, unfolding was measured. Fig. 1d shows the data for 10 cells with engaged virus particles. A closer reading of the Supplementary material showed, that only 114 events from 1073 virus particles are part of these data points. Therefore, the statistical values, which are given in the figure legend are misleading and should be better explained.

We explained the statistics in the figure legend in greater detail now, to emphasize that the data points represent a ratio of fluorescence events per available particles and note in the text that the rates represent the ensemble average of single events per virus (line 116-117 and 126-128).

For the background signal only 36 and 39 data points were measured from 9 and 11 cells, respectively. With such a low number of events per cell it is not representative to calculate mean values for each cell.

We acquired two additional datasets and analyzed three times more “random” spots not colocalizing with virus particles per cell now to increase the number of non-specific events to 88 out of 2326 random spots underneath 11 cells and 80 out of 2713 random spots in 13 control areas next to the cells. Since these background signals are non-specific events such a low probability is expected and consistently few events per cell/control region were detected.

The authors should also explain why the supplementary Movie 1 shows a decrease in signal over time.

We noted that there is significant photobleaching due to a low degree of labeling of the viruses in the figure legend and now also incorporated this in the main text (line 110-112).

It remains unclear whether the force per virus particle can be deduced from this data. Since this point is of central importance for the paper the authors should explain in greater detail how the number of sensors interacting with each virus particle can be resolved with their approach. The authors argue, that multiple sensors increase the mechanical resistance and thus 40 pN is a lower estimate for the force. However, the fluorescence signal from which the average unfolding rate is calculated, might also be caused by two or more uncorrelated sensor modules. A higher number of engaged sensor modules per particles could for example be caused by the spatial confinement of viruses below the cell, which might favor binding to multiple sensors.

Indeed, as discussed in the conclusions, we could not resolve the site-specific number of sensors per virus particle in the current study (line 470-471). There is the possibility that

multiple sensors bind one virus particle underneath the cells as well as in the control region outside cells. However, the rate for fluorescent signals at virus sites in the control region is in the same range as non-specific events occurring without viruses (these are single molecules by default, because there is no link between them) and we thus assume the signal stems mainly from singly bound viruses while multiple bonds would dramatically increase the mechanical resistance. We emphasized this point in the revised text (line 158-159).

In addition, tension is inherently overestimated, as tension sensors, which do not show a signal in the observation period of 10 min, are not included in the calculation. With an average rate of $k_{\text{off}} = 1.3 \times 10^{-4} \text{ s}^{-1}$ only 1.6 events are expected in each 10 minutes interval! For a better understanding of the data analysis procedure the number of unfolding/refolding events per time trace would be recommended.

With $k_{\text{off}} = 1.3 \times 10^{-4} \text{ s}^{-1}$ we expect 0.078 events / 10 min per virus. To overcome the detection problem of these low frequency events we did not assess the rate of unfolding/folding per individual virus particle but rather of the ensemble of total particles underneath the cell and hence also considering these tension probes, which do not show a signal. With an average of ~200 viruses per cell we thus obtained ~16 specific events per 10 min. We think this is a reasonable time interval to infer the average ensemble rates and we explained this now in more detail in the manuscript (line 117-118 and 126-128).

Ultimately the additional use of either very stiff or very soft sensor modules as well as tension sensors with a high or low zero-force unfolding rate would be recommended. However, this suggestion goes beyond the scope of this manuscript.

We thank the reviewer for this suggestions and performed initial tests with DNA-hairpin tension sensors, which require a lower unfolding force (Liu *et al.*, 2016 PNAS). However, there were too many non-specific signals of the DNA probes in the single molecule fluorescent imaging and the experiments were thus not pursued.

Part II: Uptake of biotin-neutravidin tethered virus particles and nanoparticles

In the next step, the authors proceed to a set-up where virus particles and nanoparticles are no longer covalently linked to the surface but can be taken up by the cell upon rupture of the biotin-neutravidin link. Comparing virus particles with nanoparticles coated with different ligands is a good approach to dissect biophysical mechanisms behind virus uptake. The term “virus-like” nanoparticles, which is used in the title is confusing and it would be better to call them “virus-sized”.

In addition to being of similar size, we decorated the nanoparticles with specific ligands mimicking those on the viral capsid. To avoid confusion, we simplified the title to “Forces during cellular uptake of viruses and nanoparticles at the ventral side”.

The amount of virus particles directly below the cell area decreases over time compared to background densities. This is clearly demonstrated by the data and the uptake is faster in lamellipodial regions. From the relative k_{off} rates of background and cell area, an average force on the virus particles is calculated using the Bell model. This should indeed give an approximation of the forces involved in virus particle uptake.

Next, the nanoparticles with and without specific ligands are characterized. It remains unclear why the background off-rate, which is used for the double-exponential fit of data from nanoparticles was not taken into account for the fit of virus particles.

We originally analyzed the uptake of biotin-bound virus particles analogous to the titin data in a 10 min time-interval and additionally analogous to the nanoparticle uptake considering the background off-rate. To improve clarity, we immediately present the data originating from the two-phase exponential fit in the revised manuscript (line 218-228).

It also remains to be discussed why the apparent off-rate in absence of cells is different for virus particles and nanoparticles ($k_{\text{off Virus}} = 7.2 \times 10^{-6} \text{ s}^{-1}$ vs. $k_{\text{off AuNP}} = 12.2 \times 10^{-6} \text{ s}^{-1}$) and how this influences comparison of relative forces between them.

We thank the reviewer for stressing this point. We quantified the apparent off-rate for virus particles for 7 additional data sets with a total of 6892 viruses and observed a higher statistical variance than initially revealed (illustrated in Fig. 2e). While differences between the dissociation of biotin-immobilized nanoparticles and viruses can arise from the slightly different size or binding efficiency of their biotin-linkers, we now found a similar $k_{\text{off},0}$ (virus) = $12 (\pm 3) \times 10^{-6} \text{ s}^{-1}$ as for nanoparticles. We revised the analysis of the uptake kinetics with the two-phase exponential fits accordingly, which led to a slight difference in the minimal force estimates (34 ± 2 instead of 37 ± 1 pN for HeLa and 31 ± 3 instead of 35 ± 3 pN U373 cells). Note, that we always compare the off rates induced by the cells relative to these specific background rates, which in principle allows the comparison of relative forces between different particles regardless of their $k_{\text{off}}(F=0)$. Further, the force estimates strongly depend on the accuracy of Δx , while the ratio of the rates manifest itself in the logarithm.

From the movies it seems that a subset of the particles is also pushed away by the cell rather than being taken up. Especially for nanoparticles without a specific ligand AuNP(1) many particles are dragged by the cell. Automated tracking of particles instead of only determining their position would be necessary to optimize the analysis and either exclude these events or analyze them in more detail.

We agree with the reviewer that particles were moved by the cells in various directions and included a paragraph on the translocation of the particles in x-y in the revised manuscript (line 183-188). Unfortunately, tracking of the particles was not possible due to the low temporal resolution, which was a compromise to limit phototoxicity effects, and the limited spatial resolution in z, which does not allow to determine whether particles are below or above the cell membrane apart. Preliminary TIRF data of the particle movement, showed that the translocation in x-y was often accompanied by a reduction in fluorescence intensity, suggesting a movement in z direction. However, we cannot resolve the direction of the force with the presented methods as discussed in the outlook (line 474-476). Nonetheless, we argue that the obtained values apply regardless of the direction of the force due to the flexible surface linkage of the particles.

Another way to optimize the set-up – which is beyond the scope of the current study - could be the use of monovalent forms of avidin and a lower surface density to ensure single particle interactions.

We thank reviewer 4 for this suggestion and we will consider it for future studies.

Reviewers' Comments:

Reviewer #1:

Remarks to the Author:

The authors responded to the reviewer's comments by including big parts of the supplementary notes into the main text, adding additional data sets, and rephrasing figure legends and parts of the main text. As a result, the potential limitations of the technique and the theoretical models are now more obvious; this should allow the reader to decide how accurate the here presented force values can possibly be. Unfortunately, the manuscript is in its current form very difficult to read and the authors could have done a better job in inserting the new elements into the text and structuring the manuscript. Furthermore, I am not entirely convinced about the interpretation of newly included data on the role of actin and myosin. I strongly recommend addressing these issues before publication; please also see additional comments below.

1. The authors should check whether the additional text elements were included at the right place into the main text. For instance, introducing Fig. 3a-d before discussing Fig. 2e is quite confusing.
2. The main text needs to be structured more efficiently. For instance, the equations should be introduced in a consistent fashion. Equations could be numbered (eq.1, eq.2, etc.) and the subsequent text could refer to these numbers regularly. The main text would benefit from subheadings, and the main text as well as figure legends could be more concise.
3. The authors have included additional data into Fig. 2d. However, the presented data set seems identical to the originally provided data set. Did the authors forget to include the additional data into the new Fig.2d?
4. A new data set on the role of actin polymerization and myosin contractility is included (Fig. 5). The data indicate that the particle uptake is inhibited by the drug treatments but the authors conclude that the effect plays a minor role for the generation of mechanical forces because only the fraction of particles being internalized is reduced, while the overall rate of particle uptake is hardly affected. I apologize if I misunderstood this, but would it not be important to calculate the off-rates specifically for those particles that are actively removed? Are those off-rates still unchanged by the drug treatment?
5. It really seems that, in the absence of knowledge on the loading rates and force vectors, the I27 probe does not allow very precise force measurements. The authors should cite the Li et al 2000 paper in the main text to make the reader aware of this important limitation.

Reviewer #2:

Remarks to the Author:

Although I still feel that this method has major drawbacks in terms of a large non-specific interaction energy that is not physiological, they have largely addressed the reviewer concerns. There are no further problems that I see with the experimental findings.

Reviewer #3:

Remarks to the Author:

I have studied the revision and the author's responses to my earlier comments. The concerns that I have raised previously are satisfactorily addressed. I support the acceptance of this manuscript.

Reviewer #4:

None

Point-to-point response to referees' comments:

Reviewer #1 (Remarks to the Author):

The authors responded to the reviewer's comments by including big parts of the supplementary notes into the main text, adding additional data sets, and rephrasing figure legends and parts of the main text. As a result, the potential limitations of the technique and the theoretical models are now more obvious; this should allow the reader to decide how accurate the here presented force values can possibly be. Unfortunately, the manuscript is in its current form very difficult to read and the authors could have done a better job in inserting the new elements into the text and structuring the manuscript. Furthermore, I am not entirely convinced about the interpretation of newly included data on the role of actin and myosin. I strongly recommend addressing these issues before publication; please also see additional comments below.

We acknowledge the criticism of the reviewer and we could like to point out that in the first revision we used the same model to calculate the force values from the data in Figs. 1 – 5 in a consistent manner. Furthermore, we conducted experiments with actin/myosin inhibitors, as suggested by the reviewers, and carefully discussed these data. For the second revision, we now substantially shortened the abstract, inserted subsection headings and restructured the manuscript, which greatly facilitates the readability. We now also discuss the results of the actin-myosin inhibition experiments in more detail (see below). We provide one markup version of the manuscript in which all changes are marked in yellow and paragraphs that have been moved are marked in grey. We believe that our manuscript has now become very accessible as suggested by the reviewer.

1. The authors should check whether the additional text elements were included at the right place into the main text. For instance, introducing Fig. 3a-d before discussing Fig. 2e is quite confusing.

We thank the reviewer for bringing up this point and we have now reorganized the position of the text to be consistent with the Figures. In particular we discuss Fig. 3 now in a separate subsection line 204 – 219 and switched the position of the subfigures in Fig. 4 c-d and Supplementary Fig. 6. In general, now all subfigures are being discussed in the correct order in the manuscript.

2. The main text needs to be structured more efficiently. For instance, the equations should be introduced in a consistent fashion. Equations could be numbered (eq.1, eq.2, etc.) and the subsequent text could refer to these numbers regularly. The main text would benefit from subheadings, and the main text as well as figure legends could be more concise.

In the revised manuscript we restructured large parts of the text as follows: we divided the main text in subsections and introduced subheadings. We condensed the abstract and restructured the introduction to present all biological relevant information first (shifted paragraph line 43 – 48) and closing with a brief summary of the results and conclusions of the study (line 53 – 65). We consistently numbered and referred to the main equations avoiding unnecessary repetition (line 123, 193, 216). We omitted wordy explanations in the figure legends where possible according to the journals format requirements (line 895 – 901, 929 – 932, 952 – 956, 971 – 973, 986 – 988, 993 - 994).

3. The authors have included additional data into Fig. 2d. However, the presented data set seems identical to the originally provided data set. Did the authors forget to include the additional data into the new Fig.2d?

No, these data are included. We analyzed the detachment of virus particles in control regions outside the cells in 7 additional data sets and included this data into Fig. 2d and e. While the

difference is hard to visualize due to the scaling in Fig. 2d, we chose to display the fitting parameters of the individual data points in Fig. 2e with a logarithmic scale for this reason. The axis in Fig. 2d was chosen to match the corresponding plot in Fig. 3c, 4c, 5b and Supporting Figure 5d. The source data underlying the plots are provided as source data file.

4. A new data set on the role of actin polymerization and myosin contractility is included (Fig. 5). The data indicate that the particle uptake is inhibited by the drug treatments but the authors conclude that the effect plays a minor role for the generation of mechanical forces because only the fraction of particles being internalized is reduced, while the overall rate of particle uptake is hardly affected. I apologize if I misunderstood this, but would it not be important to calculate the off-rates specifically for those particles that are actively removed? Are those off-rates still unchanged by the drug treatment?

Indeed, this is what we do. The off rates k_{off} in Fig. 3-5 all represent the kinetics of the fast decay for the fraction a of particles being actively removed as derived from fitting with a two-phase decay function (except for the background kinetics outside of cells), so these data are always measured, as expected by the referee. The second subplot in Fig. 5d shows that these off-rates are in fact changed, but not significantly, for actomyosin inhibition. The main effect is the change in the removal fraction, as shown in the first subplot of Fig. 5d. To make this point clearer we changed the axis label and order of the graphs in the figure and noted this point in the main text and the figure legends (line 310, 929-932, 952-956, 971-973, 986-988).

5. It really seems that, in the absence of knowledge on the loading rates and force vectors, the I27 probe does not allow very precise force measurements. The authors should cite the Li et al 2000 paper in the main text to make the reader aware of this important limitation.

We agree with the reviewer and now stress the nature of the force estimate in the manuscript (line 128-130). We thank the reviewer for suggesting to cite the work of Li et al, which we now correctly inserted in the revised manuscript.

Reviewer #2 (Remarks to the Author):

Although I still feel that this method has major drawbacks in terms of a large non-specific interaction energy that is not physiological, they have largely addressed the reviewer concerns. There are no further problems that I see with the experimental findings.

We thank the reviewer for acknowledging our effort in addressing his previous concerns and are glad that she/he accepts our experimental findings.

Reviewer #3 (Remarks to the Author):

I have studied the revision and the author's responses to my earlier comments. The concerns that I have raised previously are satisfactorily addressed. I support the acceptance of this manuscript.

We are very happy that the reviewer finds our work suitable for publication and would like to thank her/him for reviewing our manuscript.

Reviewers' Comments:

Reviewer #1:

Remarks to the Author:

The authors have addressed the remaining concerns. I recommend the publication of this study.